# MACS: Multi-Agent Reinforcement Learning for Optimization of Crystal Structures

Elena Zamaraeva[1]    Christopher M. Collins[*,1]    George R. Darling[*,2]    Matthew S. Dyer[*,1,2]

Bei Peng[*,3]    Rahul Savani[*,†,4,5]    Dmytro Antypov[1]    Vladimir V. Gusev[4]    Judith Clymo[4]

Paul G. Spirakis[1,4]    Matthew J. Rosseinsky[1,2]

[1]Leverhulme Research Centre for Functional Materials Design, University of Liverpool, UK
[2]Department of Chemistry, University of Liverpool, UK
[3]School of Computer Science, University of Sheffield, UK
[4]Department of Computer Science, University of Liverpool, UK
[5]The Alan Turing Institute, London, UK

## Abstract

Geometry optimization of atomic structures is a common and crucial task in computational chemistry and materials design. Following the *learning to optimize* paradigm, we propose a new multi-agent reinforcement learning method called **M**ulti-**A**gent **C**rystal **S**tructure optimization (MACS) to address periodic crystal structure optimization. MACS treats geometry optimization as a partially observable Markov game in which atoms are agents that adjust their positions to collectively discover a stable configuration. We train MACS across various compositions of reported crystalline materials to obtain a policy that successfully optimizes structures from the training compositions as well as structures of larger sizes and unseen compositions, confirming its excellent scalability and zero-shot transferability. We benchmark our approach against a broad range of state-of-the-art optimization methods and demonstrate that MACS optimizes periodic crystal structures significantly faster, with fewer energy calculations, and the lowest failure rate. Code is available at `https://github.com/lrcfmd/macs`.

## 1    Introduction

In computational chemistry, geometry optimization of an atomic structure is the process of finding a stable arrangement of atoms, following a sequence of displacements in a 3-dimensional space until a local energy minimum is reached [39]. Within geometry optimization, our focus in this paper is on crystal structures, which are characterized by their *periodicity*. The arrangement of atoms within a crystal directly determines its physical and chemical properties; hence the optimization of the crystal structure is crucial to the discovery of new crystalline materials and finds applications in electronics, energy applications, information storage, and other domains.

In fact, an entire field within materials design, known as crystal structure prediction (CSP), focuses on the computational prediction of stable crystal structures with desirable properties for subsequent synthesis in the laboratory [53, 54]. Significant effort in a CSP workflow is focused on exploring the

---

[*]Equal contribution
[†]Corresponding author, e-mail: Rahul.Savani@liverpool.ac.uk

39th Conference on Neural Information Processing Systems (NeurIPS 2025).

potential energy surface (PES) and performing geometry optimizations on candidate structures to minimize the energy and local atomic forces. In this context, the key requirement for an optimization method is the ability to quickly produce a locally optimized equilibrium structure.

Existing approaches for geometry optimization of crystal structures include classical first- and second-order optimization methods such as the Broyden–Fletcher–Goldfarb–Shanno (BFGS) algorithm [4, 14, 16, 40] and the Conjugate Gradient [41] method, as well as methods tailored to atomic structures such as the Fast Inertial Relaxation Engine (FIRE) [3]. However, these methods often require either a significant number of steps to optimize large structures or time-consuming calculations at each optimization step. This makes the time for optimization a bottleneck in applications that require thousands of local optimization runs, such as CSP.

In this work, we utilize the *learning to optimize* (L2O) paradigm [27, 8, 45] to improve the geometry optimization of crystal structures using *multi-agent reinforcement learning* (MARL). We observe that overall structural stability depends on the forces acting on individual atoms. The forces are partial derivatives (or gradients) of the energy with respect to atomic positions, and all forces are zero at equilibrium where the structure achieves a local energy minimum. Both the energy and the forces are mostly determined by each atom's chemical properties and its surrounding local environment. Moreover, atoms of a given element, as well as those of chemically related elements, tend to adopt similar chemical local environments, resulting in multiple similar local environments in an optimized crystal structure. Hence, it is natural to consider individual atoms as agents, moving independently but simultaneously to collectively discover an overall structure that is stable, i.e., a local minimum of the energy landscape. We therefore formulate crystal structure geometry optimization as a MARL problem, with the aim of learning decentralized policies for individual atoms to collectively discover a locally optimized structure.

The energy (and most often forces) estimation is integrated in any optimization method, and various methods exist for this purpose. The available methods for computing energy/forces vary in their complexity and accuracy and, therefore, cost to compute. Lennard-Jones potentials (LJ) [26, 13] or the Müller-Brown surface [37, 35] are often used to model simple systems, while more complex methods, such as density functional theory (DFT), can estimate energies and forces for materials with high accuracy but at a very much greater computational cost. We utilize CHGNet [12], a machine learning interatomic potential model trained on DFT data, as it offers fast and accurate energy and gradient estimates for crystal structures.

In this work, we make the following contributions:

- To the best of our knowledge, we are the first to apply MARL to address periodic crystal structure optimization. Our proposed method, **M**ulti-**A**gent **C**rystal **S**tructure optimization (MACS), presents a novel formulation of periodic crystal structure optimization as a multi-agent coordination problem.

- Our extensive experiments demonstrate that MACS optimizes the crystal structures significantly more efficiently than a wide range of state-of-the-art methods. These experiments cover a diverse set of crystalline materials, including compositions with different elemental species, varying numbers of species, and distinct symmetry groups.

- MACS exhibits strong zero-shot transferability and scalability, maintaining efficiency in the optimization of larger structures from new, unseen compositions. Our work unlocks the potential of MARL for periodic crystal structure optimization.

## 2   Related Work

The L2O concept leverages machine learning to develop new optimization methods tailored to specific problems, and its application is rapidly expanding. This paradigm has been successfully applied to classical optimization challenges such as Bayesian swarm optimization [6], black-box optimization [9, 28], adversarial training [55], or partial differential equations solving [18].

The geometry optimization of atomic structures is also a target of the L2O concept. In [2], the authors propose a graph-based L2O approach to optimization of finite, non-periodic atomic clusters using the LJ and Calcium silicate hydrate potentials [33], as well as the Stillinger-Weber potentials (SW) [44]. The authors show that their method achieves lower energy in optimized clusters compared to FIRE,

Adam [25], and Gradient Descent [43]. Another study [34] investigates the optimization of atomic clusters with LJ, SW, and Gupta potentials [19], focusing on the minimization of energy in the cluster.

Molecular optimization tasks, when the main objective is to achieve stable molecule configurations in the least number of steps, are explored in [36, 7, 1, 46]. In [36], the authors utilize MARL to train the MolOpt optimizer and benchmark it against three baselines: BFGS, FIRE, and MDMin [23]. The findings in [36] indicate that the MolOpt optimizer surpasses MDMin, exhibits performance comparable to FIRE, and is inferior to that of BFGS. Despite differences in the design of the Markov Decision Process (MDP) and the application to distinct classes of chemical systems, we have added all baselines of [36] in our study to maintain consistency. In [46], the authors follow another approach, where using a fixed optimizer (a variation of BFGS) they train the machine-learning potentials to be more accurate during optimization.

Reinforcement learning (RL) has also shown promise in other areas of computational chemistry, including the design of materials with specific properties [17, 24] and the optimization of the basin-hopping routine in CSP [57]. For a comprehensive overview of other applications of RL in chemistry, we direct the reader to the review in [42].

## 3 Preliminaries and Problem Formulation

### 3.1 Periodic Crystal Structures and Their Optimization

A crystal structure is characterized by its unit cell, typically, a parallelepiped, and the configuration of atoms within it. The unit cell repeats itself in all three dimensions, defining the infinite periodic arrangement of atoms (see Fig. 2a for a two-dimensional example).

Geometry or local optimization takes an initial structure as input and adjusts the positions of the atoms in the unit cell to achieve a structure where the energy is at a local minimum. An efficient procedure to perform geometry optimization is crucial due to its extensive usage throughout computational chemistry. Atomic configurations at a local minimum on the potential energy surface (PES) represent physically stable structures of a material, and thus the properties of a material commonly depend on these structures. Therefore, calculations of optoelectronic, vibrational, mechanical, and energetic properties will begin with geometry optimization to achieve a local minimum.

In the scope of crystal structure optimization, global and local optimization are distinguished by their targets. The global crystal structure optimization, which is the ultimate goal of CSP, aims to identify the global minimum energy structure for a given composition, representing the most stable configuration. However, achieving a globally optimal structure typically requires many iterations of structure generation or perturbation followed by local optimization [5, 56, 38, 49].

Geometry optimization terminates under two conditions. The first condition, termed the condition of success, requires the norm[1] of the maximum atomic forces within the structure to reach a specified threshold. We use the threshold 0.05 eV/Å, as it represents a typical use case in CSP applications [11]. The second condition, termed the condition of failure, occurs when the maximum allowable number of optimization steps is reached without satisfying the condition of success. We set the maximum number of steps to 1000, which is generally sufficient to optimize the structures within this work using state-of-the-art methods such as BFGS, BFGS with line search, or FIRE. Therefore, the problem of geometry optimization in our formulation is as follows.

> **Problem (geometry optimization):** Given an initial crystal structure and the maximum of 1000 steps, to autonomously adjust the positions of the atoms in the fixed unit cell to locally minimize all atomic forces in the structure to below 0.05 eV/Å as quickly as possible.

### 3.2 Geometry Optimization as a Partially Observable Markov Game

We model geometry optimization of crystal structures as a partially observable Markov game (POMG) [32, 31]. POMG is a multi-agent extension of MDP, with partial observability intro-

---

[1]Throughout this paper, all vectors and atomic positions are expressed in Cartesian coordinates, with distances and vector norms defined by the L2 norm.

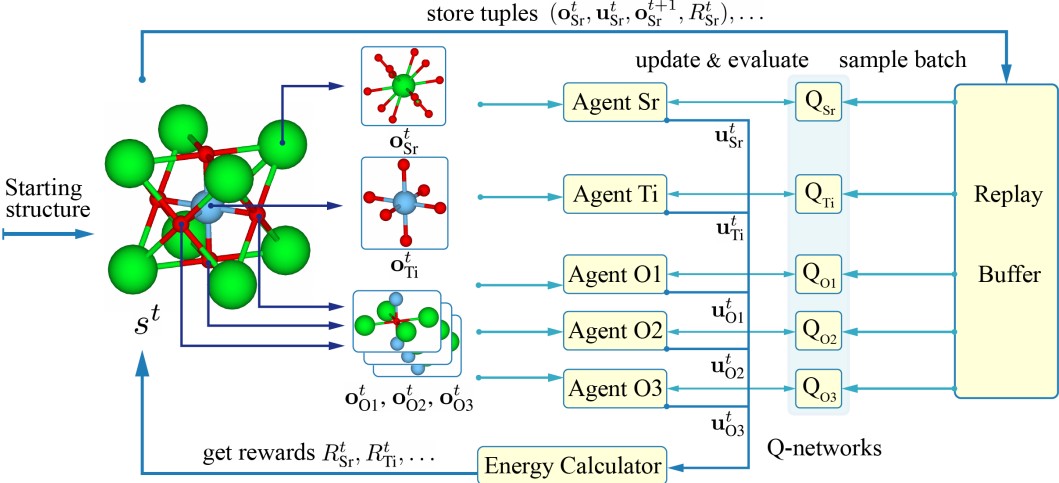

Figure 1: Our overall MACS architecture. We use the $SrTiO_3$ example with five atoms: one atom of Sr, one atom of Ti, and three atoms of O. At each time step $t$, the current state $s^t$ is converted into individual observations $o_{Sr}^t, o_{Ti}^t, o_{O1}^t, o_{O2}^t, o_{O3}^t$ that are passed to the agent (policy) networks. The agent networks output individual actions, which are then scaled and used as the atoms' displacements to update the structure. The energy calculator (CHGNet) provides the gradients for the updated structure to construct the next state $s^{t+1}$ and compute the individual rewards. The policy training process follows the standard SAC [21] workflow with policy networks, Q-networks, and replay buffer.

duced for each agent. This means that each agent has a private local observation of the global state of the environment. POMG can be represented by a tuple $<\mathbb{A}, \mathbb{S}, \mathbb{O}, \mathbb{U}, T, R_i>$, where $\mathbb{A}$ is a set of $N$ agents; $\mathbb{S}$ is the state space; $\mathbb{O} = \mathbb{O}_1 \times \cdots \times \mathbb{O}_N$ is the joint observation space, where $\mathbb{O}_i$ is the observation space of agent $a_i$; $\mathbb{U} = \mathbb{U}_1 \times \cdots \times \mathbb{U}_N$ is the joint action space, where $\mathbb{U}_i$ is the action space of $a_i$; $R_i$ is the individual reward function that returns a scalar value to agent $a_i$ for a transition from state $s \in \mathbb{S}$ to state $s' \in \mathbb{S}$ after taking joint action $u \in \mathbb{U}$; $T(s, u) : \mathbb{S} \times \mathbb{U} \to \mathbb{S}$ is the transition function, which determines the probability of transitioning to the next state $s' \in \mathbb{S}$ given that agents take joint action $u \in \mathbb{U}$ in state $s \in \mathbb{S}$. The transition function is deterministic in our problem setting.

By modelling the geometry optimization of crystal structures as a POMG, we treat each atom within a periodic unit cell of a structure as an individual agent, each with access to only local observations. All agents act independently and simultaneously to collectively discover a local minimum energy structure by maximizing their individual rewards. While each agent has a different reward function, they are aligned toward a common objective as optimizing the position of one atom improves the relative positions of the surrounding atoms. We impose partial observability intentionally to make the learning problem more tractable through the reasonable size of local observation spaces and to improve scalability to large numbers of atoms. The general scheme of MACS is presented in Fig. 1.

## 4   Methodology

In this section, we formally introduce the proposed formulation of periodic crystal structure optimization as a POMG through defining the observation space, action space, and reward function. Then we discuss our choice of the specific RL algorithm we use and its configuration.

**Observations.** Each atom in a crystal structure is surrounded by the neighboring atoms in the same unit cell, as well as their periodic images from the neighboring unit cells. Given this, we design the observation space so that each agent can observe its own features and the features of its $k$ *nearest neighbors*. The $k$ nearest neighbors refer to the $k$ atoms closest to the agent and enumerated in the order of increasing distance, either within its unit cell or from their periodic images in the neighboring unit cells. Fig. 2a shows a two-dimensional example of $k$ nearest neighbors of a specific atom in a structure.

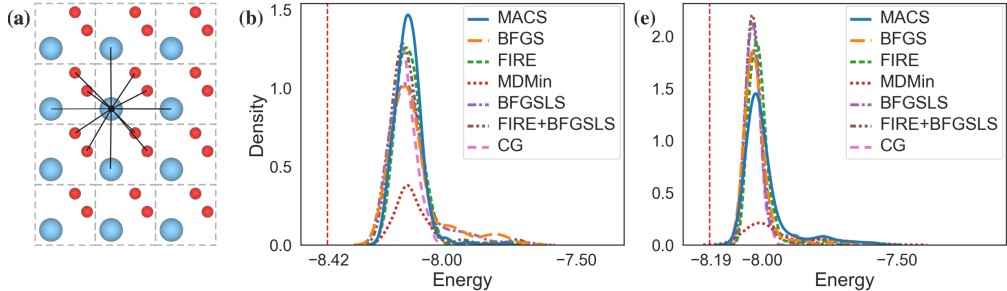

Figure 2: (a) Unit cells and nearest neighbors in two dimensions for a structure with three atoms. One atom is shown connected to its 12 nearest neighbors that belong to different unit cells; (b,c) The energy distribution of the local minima for the test sets of (b) $SrTiO_3$ with 80 atoms and (c) $Ca_2Ti_3O_7$ with 96 atoms. The vertical line indicates the energy of the experimental structure.

To define the observation space, we start with defining the feature vector $\mathbf{f}_{a_i}^t$ of agent $a_i$ at time step $t$:

$$\mathbf{f}_{a_i}^t = concat([r_i, c_i^t, \log(|\mathbf{g}_i^t|)], \mathbf{g}_i^t, \mathbf{d}_i^{t-1}, \mathbf{g}_i^t - \mathbf{g}_i^{t-1}). \tag{1}$$

Here *concat* is the concatenation function; $r_i$ is the covalent radius of $a_i$, which does not change during optimization and serves both to distinguish between different species and to carry important chemical information. The component $c_i^t$ is the action scaling factor that will be explained when defining the action space. $\mathbf{g}_i^t$ is the gradient vector of agent $a_i$ at time step $t$, which is provided by CHGNet and scaled to avoid excessively large values (see Appendix A.3); $\log(|\mathbf{g}_i^t|)$ is used in the reward function and added to the feature vector to better capture the dependence of the reward on the observation. We call by *force-related* the features that use the gradient vector that directly indicates the stability of an atom and its local environment. $\mathbf{g}_i^t - \mathbf{g}_i^{t-1}$ is the change in the gradient vector from the previous time step $t$ that reflects how successful the previous step was, and $\mathbf{d}_i^{t-1}$ is the displacement of $a_i$ at the previous time step $t-1$; these are *history* features designed to reflect the optimization dynamics. The force-related and history features prove their significance for the policy efficiency in Section 5.4.

Now, we are ready to present the full observation of $a_i$ at time step $t$ as follows:

$$\mathbf{o}_{a_i}^t = concat(\mathbf{f}_{a_i}^t, \mathbf{f}_{n_{i1}^t}^t, \ldots, \mathbf{f}_{n_{ik}^t}^t, [|\mathbf{r}_{i1}^t|, \ldots, |\mathbf{r}_{ik}^t|], \mathbf{r}_{i1}^t, \ldots, \mathbf{r}_{ik}^t). \tag{2}$$

Here, at time step $t$, $\mathbf{r}_{i1}^t, \ldots, \mathbf{r}_{ik}^t$ are the relative (with respect to $a_i$) positions of $k$ nearest neighbors of $a_i$, and $n_{i1}^t, \ldots, n_{ik}^t$ are the agents occupying those positions. Relative positions of the agent's nearest neighbors and their feature vectors reflect the geometry and chemical properties of the agent's local environment. In this study, we use $k = 12$; therefore, the observation vector has length 204. The observation design is illustrated by an example in Appendix A.1.

**Actions.** A straightforward design for the action space would be to define actions as the agent's displacements. To ensure efficient learning, we propose using a scaling factor $c_i^t$ that depends on the gradient vector norm (gnorm) of $a_i$ at time step $t$ to guide the order of magnitude of the action:

$$c_i^t = \min(|\mathbf{g}_i^t|, c_{max}). \tag{3}$$

Here $c_{max}$ is a tunable hyperparameter (in this study, $c_{max} = 0.4$) to avoid excessively large steps. Given an action $\mathbf{u}_{a_i}^t \in [-1, 1]^3$ at time step $t$, the displacement of $a_i$ is as follows:

$$\mathbf{d}_i^t = c_i^t \mathbf{u}_{a_i}^t. \tag{4}$$

The displacement can move an agent across the boundaries of the unit cell, in which case its position within the unit cell changes drastically. It is crucial that neither observation vectors nor action vectors use the positions of atoms within the unit cell, as this approach allows smooth crossing between unit cells and the handling of multiple neighbors corresponding to the same agent from different unit cells.

**Rewards.** The reward function should reflect the objective of reducing the atomic forces in a structure to sufficiently low values. Hence, we propose using the gnorms of $a_i$ at the current timestep $t$ and the next timestep $t+1$ to construct the individual reward for $a_i$ as follows:

$$R_{a_i}^t = \log(|\mathbf{g}_i^t|) - \log(|\mathbf{g}_i^{t+1}|). \tag{5}$$

The logarithmic term helps to avoid skewing the policy toward short-term gains in the early stages of optimization, given the significant difference in gnorms' magnitudes between non-optimized and optimized structures. We define an episode as the complete optimization of a structure and use the discount factor $\gamma = 0.995$ to encourage the policy to achieve the low forces more quickly.

**Independent SAC.** To train agents to collectively discover a locally optimized structure, we use independent Soft Actor-Critic (SAC), which extends SAC [21] from the single-agent to the multi-agent setting by treating all other agents as part of the environment. Hence, the multi-agent problem is decomposed into a collection of simultaneous single-agent problems that share the same environment. SAC is chosen for its sample efficiency and its use of entropy regularization, which helps prevent early convergence to suboptimal policies and promotes a balance between exploration and exploitation.

The MACS workflow with independent SAC is presented in Fig. 1. Given a structure, CHGNet is used to estimate the forces acting on the atoms. First, the gradient vectors are scaled, and then the local observation vectors are constructed and normalized. Although it may seem redundant to scale gradient vectors before normalization in the observations, this helps to preserve the gradient directions better and makes the training stable. The observation vectors are then passed on to the policy network. We use a standard SAC architecture proposed in [21] and implemented in RLlib [29], and utilize the policy network and the twin Q-networks shared between all agents for efficient training. The policy and Q-networks are two-layered MLPs with ReLU activation functions. The policy network outputs three pairs (mean, std) for the action vector, which are passed through the tanh squashing to match the action space limits. The tuples $<\mathbf{o}_{a_i}^t, \mathbf{u}_{a_i}^t, \mathbf{o}_{a_i}^{t+1}, R_{a_i}^t>$ are stored in a replay buffer with a capacity of 10 million. The hyperparameter tuning details are provided in Appendix A.

## 5 Experiments

In this section, we benchmark MACS against a set of methods commonly used for geometry optimization. We train MACS across a diverse set of chemical systems and compare its performance against baselines using various evaluation metrics that assess the efficiency and reliability of the approach. We demonstrate the scalability and zero-shot transferability of our approach by applying the trained policy to optimize structures of unseen (larger) sizes within unseen compositions. Furthermore, we conduct ablation studies to investigate the influence of our observation space, action space, and reward function designs on the performance of MACS. Finally, we compare the results of optimization by MACS and the baselines through analyzing the energy distribution for the optimized structures.

### 5.1 Training and Testing Dataset Generation

We train MACS on a set of six diverse chemical compositions: $Y_2O_3$ [57], $Cu_{28}S_{16}$ [11], $SrTiO_3$ [10], $Ca_3Ti_2O_7$ [11], $Ca_3Al_2Si_3O_{12}$ [20], and $K_3Fe_5F_{15}$ [22]. These compositions vary in the number of elements (2–4) and in the number of atoms (5–80) required to describe their experimental structures. We generate training and testing structures using the Ab Initio Random Structure Searching package (AIRSS) [30]. During training, the initial pseudo random structures are generated on the fly with the condition of belonging to one of the training compositions with equal probability and having $\sim 40$ atoms with a reasonable volume (see Appendix B.4 for more details). For every composition on which the policy is trained, we generate three test sets of 300 structures each, with the structures containing K, 1.5K, and 2K atoms, where K is the size of the structures used during training.

To demonstrate the transferability of MACS, we generate test sets for three new compositions that do not participate in the training process. Specifically, we select a composition from the training list, $SrTiO_3$, and three from the same set of elements: $Sr_2TiO_4$, $Sr_3Ti_2O_7$, and $Sr_4Ti_3O_{10}$. For each of these new compositions, we create two test sets: one with structures approximately the same size as training structures and the other with structures twice the size.

### 5.2 Baselines and Evaluation Metrics

We benchmark MACS against six baselines: **BFGS** is a quasi-Newton method that approximates the Hessian matrix based on gradient information; **BFGSLS** is a variation of BFGS with line search [52]; **FIRE** is a first-order method which is based on the molecular dynamics approach with additional velocity adjustments and adaptive time steps; **FIRE+BFGSLS** is a hybrid approach where up to

250 steps of FIRE are followed by up to 750 steps of BFGSLS to fine-tune the structure [15, 11]; **MDMin** is a modification of the velocity-Verlet method with all masses of atoms equal to 1; **CG** stands for the conjugate gradient baseline, specifically, the Polak-Ribiere algorithm. All baselines are implemented in either the Atomic Simulation Environment package (ASE) [23] or SciPy [48]. We allow the optimization of a structure to last up to 1000 steps or until the forces of all atoms are below the defined threshold.

A natural way to compare geometry optimization methods is by the time and number of steps required to optimize an initial configuration, hence our first two evaluation metrics are the mean number of steps among successful optimizations ($\mathbf{N}_{mean}$) and the mean optimization time ($\mathbf{T}_{mean}$), while the contribution of the training time is discussed separately in Appendix B.2. We observe that for BFGSLS and CG, the number of steps is not equal to the number of energy calculations, as each step in both algorithms can involve more than one energy calculation. As energy calculations contribute significantly to the optimization time, we also use the mean number of energy calculations among the successful optimizations ($\mathbf{C}_{mean}$) as an evaluation metric. Finally, we take into account the failure rate ($\mathbf{P}_F$) of each method to estimate their reliability.

Table 1: Performance comparison of MACS and the baselines on all test sets, covering $\mathbf{T}_{mean}$, $\mathbf{C}_{mean}$, $\mathbf{N}_{mean}$, and $\mathbf{P}_F$. The metrics $\mathbf{N}_{mean}$ and $\mathbf{C}_{mean}$ are presented either as a single value (if they are identical) or separated by a ';' symbol (if they differ). The lowest values of $\mathbf{T}_{mean}$, $\mathbf{C}_{mean}$, and $\mathbf{P}_F$ are shown in bold. The standard errors are provided in Appendix B.6.

| Composition | N atoms | $\mathbf{T}_{mean}$ (sec) | | | | | | | $\mathbf{N}_{mean}$ ; $\mathbf{C}_{mean}$ | | | | | | |
|---|---|---|---|---|---|---|---|---|---|---|---|---|---|---|---|
| | | MACS | BFGS | FIRE | MDMin | BFGSLS | FIRE+BFGSLS | CG | MACS | BFGS | FIRE | MDMin | BFGSLS | FIRE+BFGSLS | CG |
| $Y_2O_3$ | 40 | **18** | 48 | 42 | 74 | 28 | 38 | 64 | **121** | 313 | 262 | 442 | 137 ; 185 | 252 ; 267 | 122 ; 543 |
| | 60 | **32** | 85 | 92 | 155 | 70 | 74 | 118 | **147** | 340 | 338 | 553 | 178 ; 281 | 324 ; 357 | 145 ; 642 |
| | 80 | **48** | 137 | 130 | 207 | 97 | 112 | 184 | **169** | 395 | 393 | 625 | 206 ; 307 | 360 ; 403 | 171 ; 754 |
| $Cu_{28}S_{16}$ | 44 | **29** | 62 | 47 | 116 | 54 | 47 | 81 | **150** | 293 | 257 | 543 | 147 ; 177 | 232 ; 242 | 158 ; 716 |
| | 66 | **51** | 112 | 88 | 205 | 60 | 79 | 147 | **186** | 355 | 307 | 633 | 176 ; 201 | 280 ; 291 | 198 ; 881 |
| | 88 | **74** | 175 | 120 | 315 | 110 | 117 | 275 | **230** | 414 | 392 | 745 | 213 ; 239 | 352 ; 365 | 283 ; 1269 |
| $SrTiO_3$ | 40 | **57** | 94 | 109 | 248 | 65 | 102 | 134 | **143** | 255 | 314 | 625 | 133 ; 190 | 276 ; 299 | 141 ; 572 |
| | 60 | **90** | 163 | 202 | 461 | 138 | 200 | 250 | **179** | 316 | 379 | 719 | 169 ; 255 | 321 ; 406 | 168 ; 681 |
| | 80 | **142** | 242 | 366 | 672 | 214 | 332 | 452 | **208** | 329 | 446 | 765 | 199 ; 317 | 355 ; 433 | 205 ; 837 |
| $Ca_3Ti_2O_7$ | 48 | **59** | 135 | 119 | 264 | 68 | 120 | 156 | **146** | 270 | 324 | 623 | 136 ; 185 | 284 ; 317 | 151 ; 618 |
| | 72 | **106** | 199 | 239 | 479 | 184 | 252 | 301 | **183** | 310 | 408 | 707 | 168 ; 249 | 335 ; 374 | 186 ; 756 |
| | 96 | **163** | 276 | 412 | 705 | 195 | 324 | 499 | **205** | 353 | 467 | 762 | 193 ; 267 | 369 ; 447 | 213 ; 876 |
| $K_3Fe_5F_{15}$ | 46 | **31** | 82 | 124 | 155 | 68 | 98 | 132 | **111** | 274 | 246 | 501 | 135 ; 178 | 246 ; 263 | 134 ; 602 |
| | 69 | **51** | 146 | 214 | 276 | 118 | 181 | 248 | **128** | 320 | 293 | 596 | 163 ; 259 | 299 ; 344 | 160 ; 720 |
| | 92 | **96** | 236 | 377 | 485 | 154 | 271 | 371 | **143** | 359 | 326 | 642 | 176 ; 236 | 353 ; 396 | 181 ; 815 |
| $Ca_3Al_2Si_3O_{12}$ | 40 | 117 | 127 | 226 | 585 | 167 | 190 | 307 | 209 | **189** | 307 | 700 | 141 ; 264 | 269 ; 311 | 147 ; 553 |
| | 60 | 237 | 266 | 389 | 1068 | 276 | 422 | 572 | 264 | **230** | 382 | 755 | 165 ; 296 | 316 ; 391 | 177 ; 669 |
| | 80 | 343 | **333** | 627 | 1279 | 400 | 627 | 958 | 317 | **246** | 461 | 894 | 189 ; 327 | 350 ; 458 | 214 ; 814 |
| **Compositions unseen during training** | | | | | | | | | | | | | | | |
| $Sr_2TiO_4$ | 56 | **65** | 145 | 174 | 361 | 105 | 139 | 234 | **172** | 335 | 371 | 700 | 162 ; 218 | 319 ; 353 | 175 ; 716 |
| | 112 | **189** | 358 | 665 | 759 | 414 | 440 | 559 | **245** | 420 | 554 | 850 | 242 ; 397 | 427 ; 508 | 245 ; 1003 |
| $Sr_3Ti_2O_7$ | 48 | **54** | 123 | 151 | 315 | 88 | 130 | 191 | **153** | 288 | 345 | 676 | 153 ; 210 | 299 ; 323 | 167 ; 682 |
| | 96 | **159** | 369 | 497 | 800 | 366 | 375 | 477 | **227** | 382 | 501 | 817 | 212 ; 343 | 385 ; 449 | 223 ; 909 |
| $Sr_4Ti_3O_{10}$ | 34 | **30** | 77 | 87 | 174 | 48 | 76 | 106 | **126** | 251 | 282 | 547 | 124 ; 173 | 256 ; 275 | 137 ; 557 |
| | 68 | **113** | 202 | 238 | 498 | 149 | 205 | 307 | **186** | 310 | 408 | 729 | 179 ; 304 | 339 ; 385 | 183 ; 743 |
| AVERAGE[2] | 66 | **99** | 175 | 239 | 444 | 152 | 207 | 297 | **182** | 315 | 366 | 673 | 171 ; 253 | 317 ; 361 | 179 ; 747 |
| $\mathbf{P}_F$[3] (%) | | **0.36** | 3 | 9.64 | 46.19 | **0.36** | 0.82 | 18.22 | | | | | | | |

[2] The average is taken as the average across the metric values for all tests, i.e. the mean of the column above.

[3] The average $\mathbf{P}_F$ across all compositions. $\mathbf{P}_F$ per test set is provided in Tab. 8.

## 5.3 MACS Policy Evaluation

We train MACS for $\sim$80,000 steps in total. The analysis of the variability of the policy trained starting from the different random seeds is provided in Appendix B.5 and confirms its consistent performance. After training, we optimize the structures in the test sets using MACS and the baselines on the same hardware, allowing exactly one CPU per optimization (see Appendix B.1 for more details).

Tab. 1 shows the optimization results of MACS and the baselines across all test sets, covering the four evaluation metrics ($\mathbf{T}_{mean}$, $\mathbf{N}_{mean}$, $\mathbf{C}_{mean}$, and $\mathbf{P}_F$) mentioned above. We observe that MACS is substantially faster and requires fewer energy calculations than all baselines in nearly all test sets,

with only a few exceptions. Specifically, on average, $\mathbf{T}_{\text{mean}}$ and $\mathbf{C}_{\text{mean}}$ of MACS are 34% and 28% less than those of the best baseline, BFGSLS, respectively. We can also see that MACS has the lowest failure rate ($\mathbf{P}_F = 0.36\%$) and performs comparably to BFGSLS on this metric. In terms of $\mathbf{N}_{\text{mean}}$, BFGSLS performs slightly better than MACS, requiring 5% fewer steps on average. MACS performs comparably to CG and outperforms all other baselines. However, CG has a much higher failure rate ($\mathbf{P}_F = 18.22\%$) than MACS. Both BFGSLS and CG involve multiple energy calculations per step, resulting in more total energy calculations and longer optimization time compared to MACS.

MACS consistently outperforms all baselines in $\mathbf{T}_{\text{mean}}$ and $\mathbf{C}_{\text{mean}}$ across all compositions and structure sizes, except for $Ca_3Al_2Si_3O_{12}$, where MACS ranks first or second after BFGS (see also Appendix B.8). This demonstrates the scalability of our method, as it maintains competitive performance as the structure size increases. Moreover, MACS exhibits excellent zero-shot transferability, as it outperforms all baselines in $\mathbf{T}_{\text{mean}}$ and $\mathbf{C}_{\text{mean}}$ in all sets of structures of compositions on which it was not trained.

Figs. 2b and 2c show the energy distributions for local minima obtained by different methods. We can see that, when optimizing the same set of structures, MACS and the baselines sample from the same distribution of local minima. Figs. 3a and 3d show the energy evolution averaged over all successful optimizations for the test sets of the $SrTiO_3$ and $Ca_3Al_2Si_3O_{12}$ structures containing 80 atoms. It demonstrates that MACS decreases energy faster than the baselines or performs comparably to the best ones among them.

The analysis of the optimized structures based on interatomic distances is presented in Appendix B.7.

## 5.4 Ablation Studies

We compare the MACS design proposed in Section 4 (referred to as MACS) with its modifications.

**Observations.** We perform ablation experiments to investigate the influence of the feature representation in the observation space on our method's performance. Specifically, we compare the MACS atom feature vector provided in Eq. 1 with the feature vectors in Eqs. 6 to 9 (referred to as feat.6, feat.7, feat.8, or feat.9), which use reduced feature representations that exclude some force-related or history features. For all setups, we evaluate the mean episodic reward and the mean episode length achieved by them across all compositions during training. As shown in Figs. 3b and 3e, MACS and feat.9 achieve the best performance, while feat.6 shows the worst performance. This demonstrates the importance of including force-related features in the observation space. We then evaluate the policies trained with different feature designs on optimizing the $SrTiO_3$ composition from the test sets.

Tab. 2 shows that MACS performs substantially better than feat.7 and feat.8 in $\mathbf{T}_{\text{mean}}$ and $\mathbf{N}_{\text{mean}}$. The design feat.9 shows marginally lower performance in $\mathbf{N}_{\text{mean}}$ and better performance in $\mathbf{T}_{\text{mean}}$ compared to MACS. In Appendix B.9, we further evaluate feat.9 on all remaining test sets and see that it takes on average $\sim$7.7% more optimization steps than MACS. These results

$$\mathbf{f}_{a_i}^t = concat([r_i, c_i^t], \mathbf{d}_i^{t-1}), \tag{6}$$

$$\mathbf{f}_{a_i}^t = concat([r_i, c_i^t, \log(|\mathbf{g}_i^t|)], \mathbf{g}_i^t), \tag{7}$$

$$\mathbf{f}_{a_i}^t = concat([r_i, c_i^t, \log(|\mathbf{g}_i^t|)], \mathbf{g}_i^t, \mathbf{g}_i^t - \mathbf{g}_i^{t-1}), \tag{8}$$

$$\mathbf{f}_{a_i}^t = concat([r_i, c_i^t, \log(|\mathbf{g}_i^t|)], \mathbf{g}_i^t, \mathbf{d}_i^{t-1}). \tag{9}$$

show that both force-related and history features are crucial to the competitive performance of MACS.

**Rewards.** To investigate the influence of the reward function on the performance of our method, we explore two additional reward designs. For the first reward design (rew.10), we add a fixed *penalty* (tuned to -0.05) at each step to the reward used in Eq. 5, to explore whether this encourages faster optimization:

$$R_{a_i}^t = \log(|\mathbf{g}_i^t|) - \log(|\mathbf{g}_i^{t+1}|) + penalty. \tag{10}$$

The second reward design (rew.11) explores the effects of partial reward sharing by adding the average reward across all agents to the original individual reward used in Eq. 5:

$$R_{a_i}^t = \log(|\mathbf{g}_i^t|) - \log(|\mathbf{g}_i^{t+1}|) + \frac{1}{N}\sum_{j=1}^{N}(\log(|\mathbf{g}_j^t|) - \log(|\mathbf{g}_j^{t+1}|)). \tag{11}$$

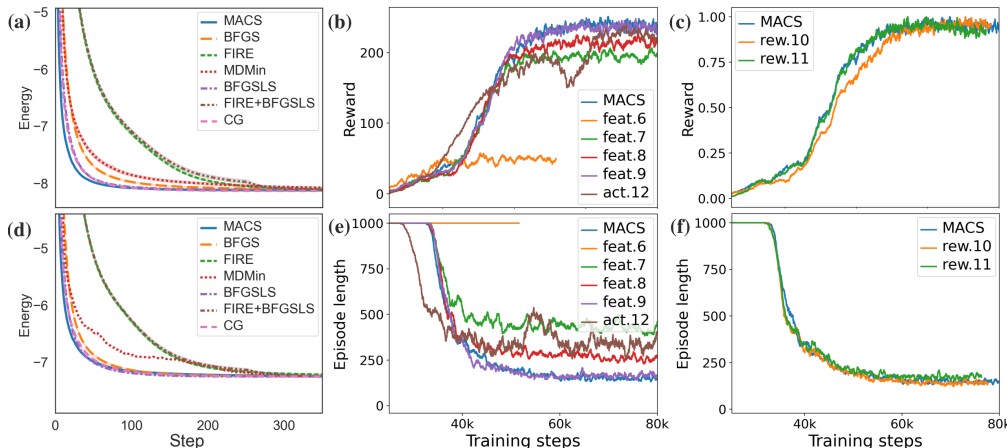

Figure 3: (a, d) Energy evolution averaged over all successfully optimized structures of 80 atoms within compositions $SrTiO_3$ (a) and $Ca_3Al_2Si_3O_{12}$ (d); (b,c,e,f) The ablation studies: the discounted episodic reward (b, e) and the mean episode length (c, f) across all compositions during training for MACS and its modifications.

Table 2: Performance comparison of MACS with varying feature representations (feat.7-9) in the observation space and varying reward functions (rew.10,11). The best numbers are shown in bold. The design feat.6 is excluded due to its inability to successfully optimize structures.

| Composition | N atoms | $\mathbf{T}_{mean}$ (sec) | | | | | | $\mathbf{N}_{mean}$ | | | | | |
|---|---|---|---|---|---|---|---|---|---|---|---|---|---|
| | | MACS | feat.7 | feat.8 | feat.9 | rew.10 | rew.11 | MACS | feat.7 | feat.8 | feat.9 | rew.10 | rew.11 |
| | 40 | 57 | 146 | 79 | **46** | 50 | 54 | **143** | 416 | 256 | 145 | 144 | 169 |
| $SrTiO_3$ | 60 | 90 | 242 | 147 | **82** | 88 | 83 | **179** | 504 | 303 | 184 | 187 | 202 |
| | 80 | 142 | 385 | 236 | **132** | 154 | 161 | **208** | 572 | 373 | 209 | 220 | 239 |
| AVERAGE | 60 | 96 | 258 | 154 | **87** | 97 | 99 | **177** | 498 | 311 | 180 | 184 | 204 |
| $\mathbf{P}_F$ (%) | | 0.22 | 8.11 | 0.44 | 0.44 | **0** | 0.33 | | | | | | |

Fig. 3c and 3f show the mean episodic reward and mean episode length achieved by our method with different reward functions. Note that the episodic reward is normalized to make performance comparable between different reward functions. We can see that rew.10 and rew.11 perform similarly to MACS. We then evaluate the policies trained with the three reward functions on optimizing the $SrTiO_3$ composition from the test sets. Tab. 2 shows that, MACS achieves superior performance than rew.10 and rew.11 in $\mathbf{T}_{mean}$ and $\mathbf{N}_{mean}$ on the test set with the largest structures (80 atoms).

**Actions.** We consider a straightforward action design (act.12) in which the action vector is used directly as atom's displacement (without the scaling factor $c_i^t$ as defined in Eq. 3):

$$\mathbf{d}_i^t = \mathbf{u}_{a_i}^t. \tag{12}$$

Fig. 3b shows that both action designs converge to similar episodic rewards, but act.12 is significantly less sample-efficient and stable. Moreover, Fig. 3e shows that act.12 converges to higher average episode length, resulting in more optimization steps. This demonstrates that the scaling factor introduced in our action design is crucial to the competitive performance of MACS, enabling more sample-efficient and stable learning.

## 6 Conclusion

In this work, we present MACS, a new MARL method for periodic crystal structure optimization. MACS introduces a novel model of geometry optimization as a multi-agent coordination problem, an unexplored direction that poses unique challenges in balancing expressive yet compact representations of chemical and geometric information, modelling complex atomic interactions, and enabling efficient, scalable policy learning. We conduct extensive experiments comparing MACS with various state-of-the-art methods and demonstrate that it learns a policy capable of efficiently optimizing crystal

structures across a diverse set of test cases, significantly outperforming all baselines. We show that on average across all test sets, MACS optimizes the structures 34% faster and with 28% fewer energy calculations than the strongest baseline, BFGSLS, preserving the lowest failure rate.

MACS demonstrates scalability and zero-shot transferability, as it is superior to the baselines in the optimization of the structures of larger sizes and compositions that it does not encounter during training. In conclusion, MACS has the potential to evolve into a universal geometric optimizer for periodic crystal structures.

**Limitations and future work**. In this study, the only feature that differentiates atomic species is the covalent radius. Various atomic features and existing descriptor implementations [58, 47] will be considered in the future. The analysis of the observation space in Section 5.4 confirmed that the history plays a crucial role in the efficiency of the method. Hence, the integration of recurrent neural networks into the method could improve MACS, and we will investigate this matter further. Another promising direction for future work is learning to optimize the unit cell of the structure. We plan to treat the unit cell vectors as separate agents, optimizing them in a manner similar to the way atoms optimize their positions. Finally, alternative approaches to estimating the energy of structures can be used in place of CHGNet. Two contrasting concepts can be considered: one involves using noisy energy and force evaluations to mitigate inaccuracies of a machine-learning interatomic potential method, while the other relies using DFT, a more accurate, but also a more computationally expensive method. As CHGNet is trained on DFT, the train-with-CHGNet-run-with-DFT workflow presents a promising concept for future study.

# 7 Acknowledgements

The authors would like to thank Professor Igor Potapov who gave valuable feedback that improved the quality of our work.

The authors acknowledge funding from the Leverhulme Trust via the Leverhulme Research Centre for Functional Materials Design (RC-2015-036) and UK Engineering and Physical Sciences Research Council (EPSRC) through grant number EP/V026887. This project has also received funding from the AI for Chemistry: AIchemy Hub (EPSRC grant EP/Y028775/1 and EP/Y028759/1). V. V. Gusev is supported by the Leverhulme Research Fellowship.

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

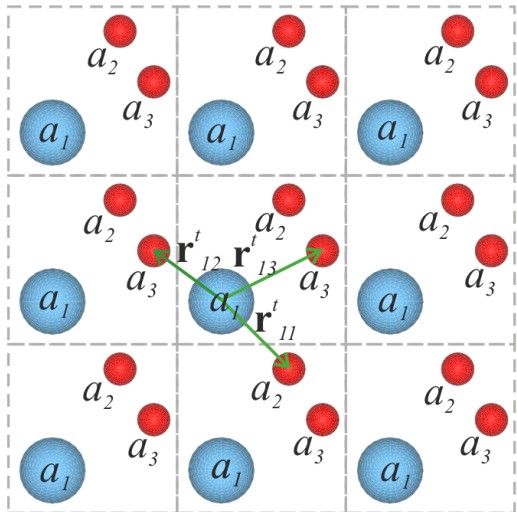

Figure 4: Two-dimensional structure with three atoms, and the three nearest neighbors for one of the atoms $a_1$.

## A The MACS design and hyperparameter tuning

### A.1 The nearest neighbors example

We consider a two-dimensional example of a structure with a square unit cell and three atoms $a_1, a_2, a_3$, where $a_2$ and $a_3$ relate to the same chemical element. We assume that the structure is undergoing optimization, and Fig. 4 shows the atomic arrangement at time step $t$. Here, for agent $a_1$, the first nearest neighbor is a periodic image of atom $a_2$ in a neighboring unit cell, hence $n_{11}^t = a_2$; the second nearest neighbor is a periodic image of $a_3$ in another neighboring unit cell, and the third nearest neighbor is atom $a_3$ itself, hence $n_{12}^t = n_{13}^t = a_3$. Furthermore, $\mathbf{r}_{11}^t$, $\mathbf{r}_{12}^t$, and $\mathbf{r}_{13}^t$ denote the positions of the three nearest neighbors of $a_1$, relative to $a_1$, i.e. vectors in Euclidean space from $a_1$ to each of its three nearest neighbors. Therefore, the observation vector for agent $a_1$ at time step $t$ looks as follows:

$$\mathbf{o}_{a_1}^t = concat(f_{a_1}^t, f_{a_2}^t, f_{a_3}^t, f_{a_3}^t, [|\mathbf{r}_{11}^t|, |\mathbf{r}_{12}^t|, |\mathbf{r}_{13}^t|], \mathbf{r}_{11}^t, \mathbf{r}_{12}^t, \mathbf{r}_{13}^t). \tag{13}$$

During the optimization process, the list of $k$ nearest neighbors is updated at each time step: existing nearest neighbors may update their relative positions and ordering in the list, while some may leave the list and are replaced by new ones.

The amd package [50, 51] is used for the fast construction of the ordered list of the $k$ nearest neighbors for all atoms on each step.

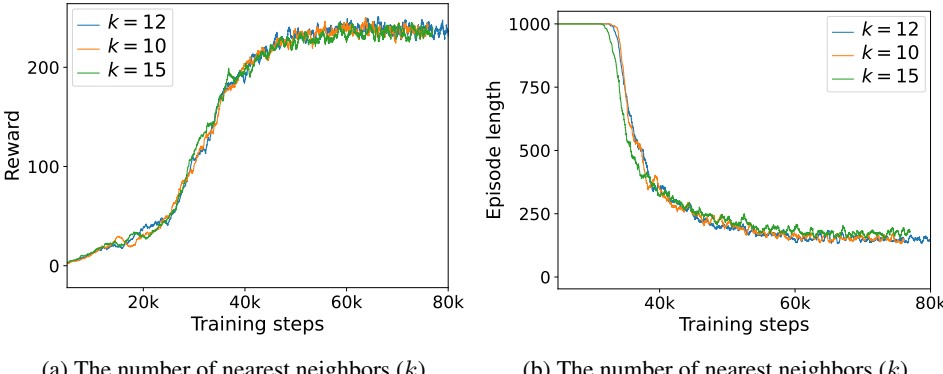

(a) The number of nearest neighbors ($k$)          (b) The number of nearest neighbors ($k$)

Figure 5: Mean episodic reward and mean episode length achieved by MACS with different numbers of nearest neighbors considered in agent's local observation.

## A.2  The number of nearest neighbors $k$

We explore the number of nearest neighbors that should be considered in the local observation of an agent. We choose 10, 12, and 15 nearest neighbors and train three policies using MACS that differ only in this hyperparameter. Fig. 5 shows that the three policies learned by MACS achieve comparable performance in both mean episodic reward and mean episode length during training. We then optimize the structures of the test sets of compositions $Y_2O_3$, $SrTiO_3$, and $Ca_3Al_2Si_3O_{12}$ with these policies. Tab. 3 shows that the policy with $k = 15$ has the highest failure rate and the longest optimization. The policies with $k = 10$ and $k = 12$ are comparable, with the latter being marginally better and therefore chosen for this study.

Table 3: Performance comparison of MACS with different numbers of nearest neighbors considered in agent's local observation. The best number is in bold, the standard errors are in brackets.

| Composition | N atoms | $\mathbf{T_{mean}}$ (sec) | | | $\mathbf{N_{mean}}$ | | |
|---|---|---|---|---|---|---|---|
| | | $k = 10$ | $k = 12$ | $k = 15$ | $k = 10$ | $k = 12$ | $k = 15$ |
| $Y_2O_3$ | 40 | **17(0)** | 18(1) | 26(2) | 123(2) | **121(3)** | 137(4) |
| | 60 | 34(1) | **32(1)** | 48(3) | 150(3) | **147(3)** | 171(5) |
| | 80 | 51(1) | **48(1)** | 81(4) | 179(4) | **169(3)** | 201(6) |
| $SrTiO_3$ | 40 | **48(1)** | 57(12) | 51(2) | 148(3) | **143(3)** | 164(4) |
| | 60 | **76(2)** | 90(12) | 88(2) | **178(4)** | 179(4) | 196(4) |
| | 80 | 139(3) | 142(13) | **138(3)** | 215(5) | **208(5)** | 232(5) |
| $Ca_3Al_2Si_3O_{12}$ | 40 | 124(4) | **117(4)** | 148(6) | **207(5)** | 209(5) | 238(7) |
| | 60 | **231(7)** | 237(14) | 245(8) | 274(7) | **264(7)** | 307(8) |
| | 80 | 380(11) | **343(16)** | 436(13) | 333(8) | **317(8)** | 367(9) |
| AVERAGE | 60 | 122 | **120** | 140 | 201 | **196** | 224 |
| $\mathbf{P}_F$ (%) | | **0.56** | 0.78 | 3.52 | | | |

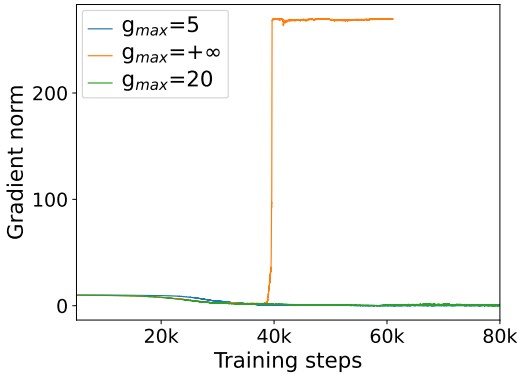

Figure 6: The gradient norm during training for MACS with different $g_{max}$.

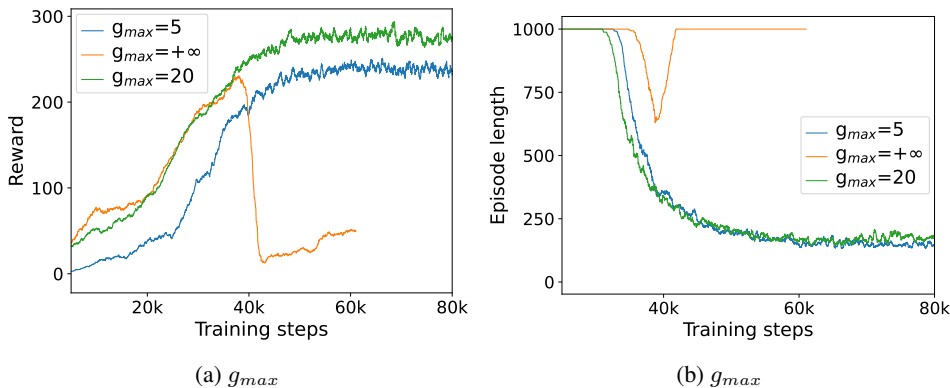

(a) $g_{max}$

(b) $g_{max}$

Figure 7: Performance comparison of MACS with different values of $g_{max}$.

### A.3 Gradient vector scaling

Let $\mathbf{go}_i^t$ be the gradient vector provided by CHGNet or any other energy/gradients calculator for agent $a_i$ at time step $t$. The corresponding scaled gradient vector used for observations/actions/rewards in our study is calculated as follows:

$$\mathbf{G}_i^t = \begin{cases} \mathbf{go}_i^t, & \text{if } \|\mathbf{go}_i^t\|_\infty < g_{max}, \\ \mathbf{go}_i^t \times \frac{g_{max}}{\|\mathbf{go}_i^t\|_\infty}, & \text{otherwise.} \end{cases} \tag{14}$$

Here $g_{max}$ is a tunable parameter. Our experiments showed that, while gradient vectors generally have components in the range $[-50, 50]$ at the beginning of optimization, occasionally there can be vectors with components up to 500. Such gradient vectors significantly unbalance training and eventually lead to gradient explosion (see Fig. 6). In practice, there is no difference between large and extremely large gradient vectors, as they all indicate a very undesirable atomic environment. Scaling the gradient vector to reasonable component values that preserve the direction helps mitigate this problem. Fig. 7 shows the mean episodic reward and the mean episode length achieved by MACS during training with different values of $g_{max}$. MACS with $g_{max} = 20$ converges to the highest reward, which is expected due to the higher gnorms at the beginning of the optimization, and hence, higher rewards. However, MACS with $g_{max} = 5$ converges to the lowest mean episode length while achieving high episodic reward, therefore, in this study, we use $g_{max} = 5$.

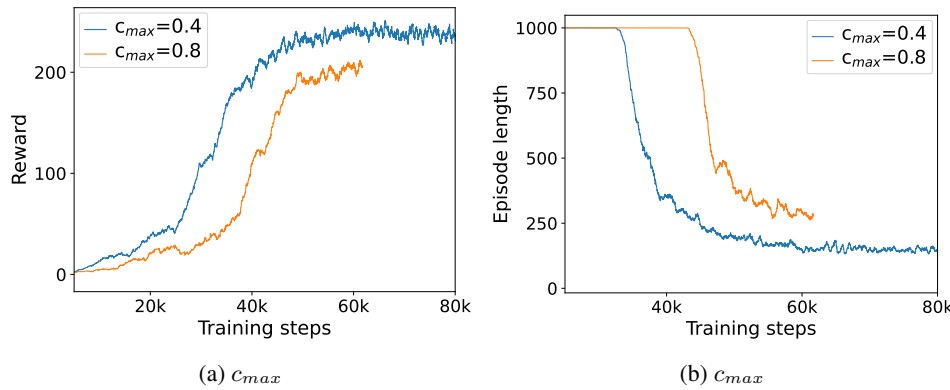

(a) $c_{max}$

(b) $c_{max}$

Figure 8: Performance comparison of MACS with different values of $c_{max}$.

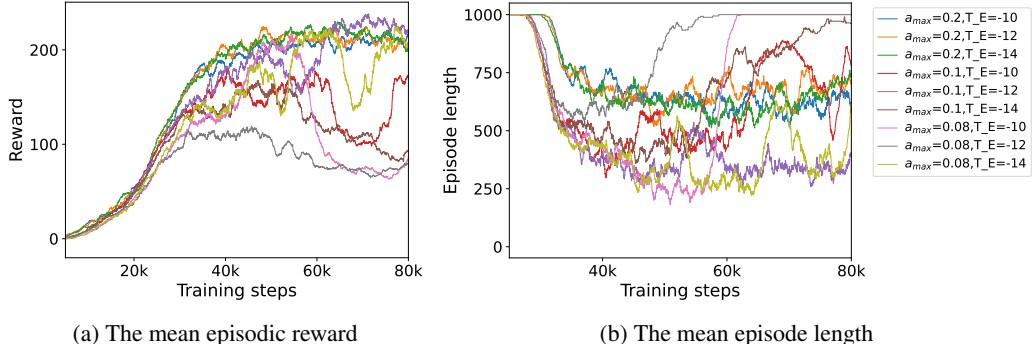

(a) The mean episodic reward

(b) The mean episode length

Figure 9: Performance comparison of MACS with different action space bounds and target entropy values.

## A.4  Hyperparameter tuning for actions

The scaling factor $c_i^t$ is applied for the action of agent $a_i$ at time step $t$ to guide the magnitude of the atom's displacement. The upper boundary for $c_i^t$ is defined by a hyperparameter $c_{max}$ to avoid excessively large displacements. Fig. 8 compares two variants of $c_{max}$ and shows that the smaller $c_{max}$ allows the policy to converge to much higher mean episodic reward and lower episode length, and thus $c_{max} = 0.4$ was chosen for this study.

We also explore a straightforward approach for designing the action space: the action vector is used directly as the atom's displacement (Eq. 12), without the scaling factor $c_i^t$. We consider different bounds on the action space to investigate its effect on policy learning, namely, we consider $[-a_{max}, a_{max}]^3$ for different values of $a_{max}$. We notice that, in SAC, the outputs of the policy network are tanh squashed and then scaled to fit the action space limits. It leads to different policy outputs' magnitudes for the same action vector, depending on the action space limits. To take this into account, we also explore different values of the target entropy. We compare the mean episodic reward and the mean episode length achieved by MACS with different action space bounds and target entropy values. Fig. 9 shows that wider bounds on action space ($a_{max} = 0.2$) make training more stable, while narrower bounds on action space ($a_{max} = 0.1$) allow for achieving a lower average episode length. We choose the best performing variant, namely, $a_{max} = 0.1$ with the target entropy T_E$= -12$ for the ablation study.

Table 4: Hyperparameters used to train MACS.

| Hyperparameter | value |
|---|---|
| $\gamma$ | 0.995 |
| Training batch size | 8192 |
| Target entropy | -8 |
| Truncate episodes | TRUE |
| Target network update frequency | 1000 |
| Number of samples before learning starts | 500 |
| Tau | 0.001 |
| Initial alpha | 1 |
| Use twin q | TRUE |
| Actor learning rate | 0.0003 |
| Critic learning rate | 0.0003 |
| Entropy learning rate | 0.0001 |
| Replay buffer capacity | 10000000 |
| Use prioritised replay buffer | FALSE |
| $g_{max}$ | 5 |
| $c_{max}$ | 0.4 |
| Observation component-wise normalization | TRUE |
| Number of nearest neighbors $k$ | 12 |
| Max steps in episode | 1000 |

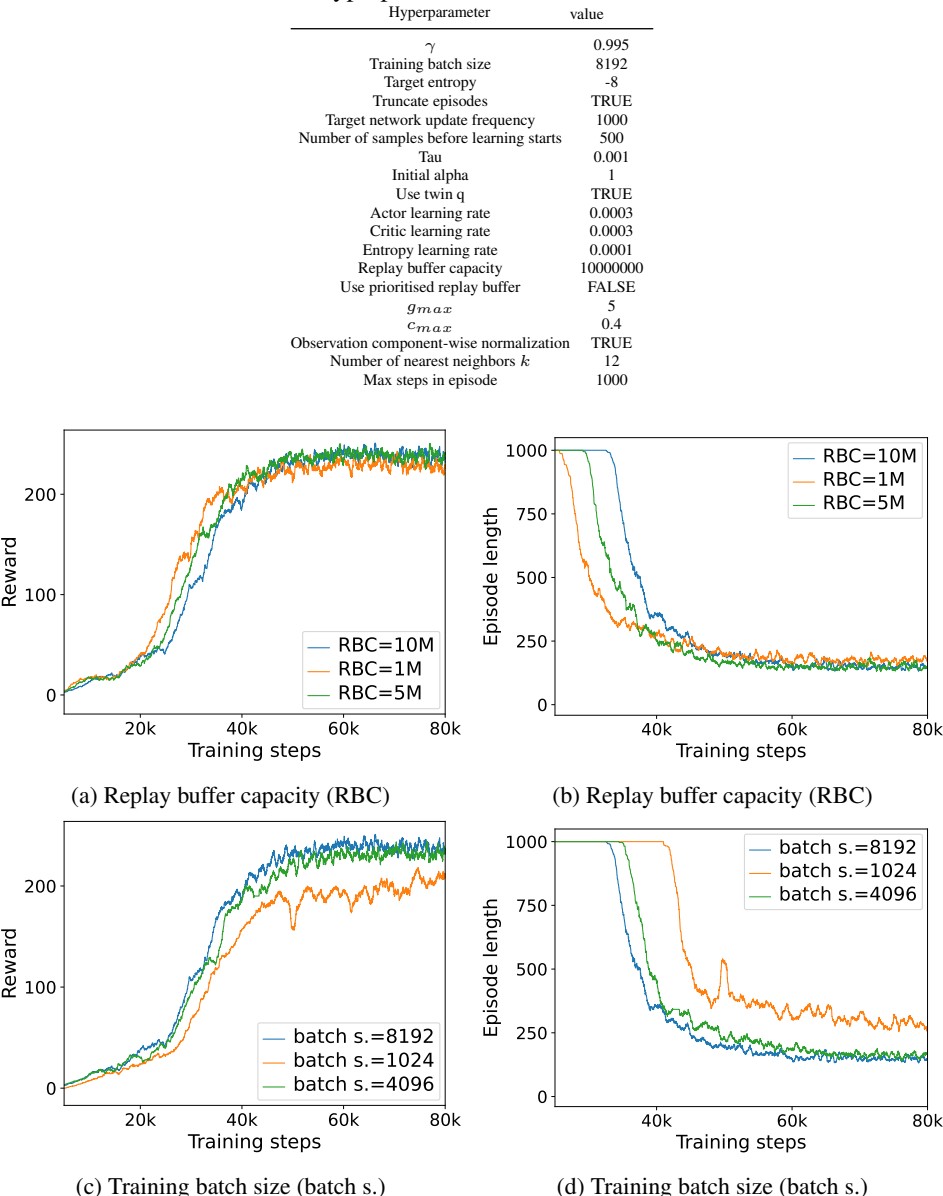

(a) Replay buffer capacity (RBC)

(b) Replay buffer capacity (RBC)

(c) Training batch size (batch s.)

(d) Training batch size (batch s.)

Figure 10: Performance comparison of MACS with different values of the replay buffer capacity and training batch size. The blue lines always indicate the values used in the paper.

## A.5  Other hyperparameters

The list of hyperparameters used to train MACS is shown in Tab. 4. Figs. 10 and 11 show the mean episodic reward and the mean episode length achieved by MACS during training for different variations of the hyperparameters. We can see that the training batch size (Figs. 10c and 10d) and the entropy learning rate (Figs. 11c and 11d) play a crucial role.

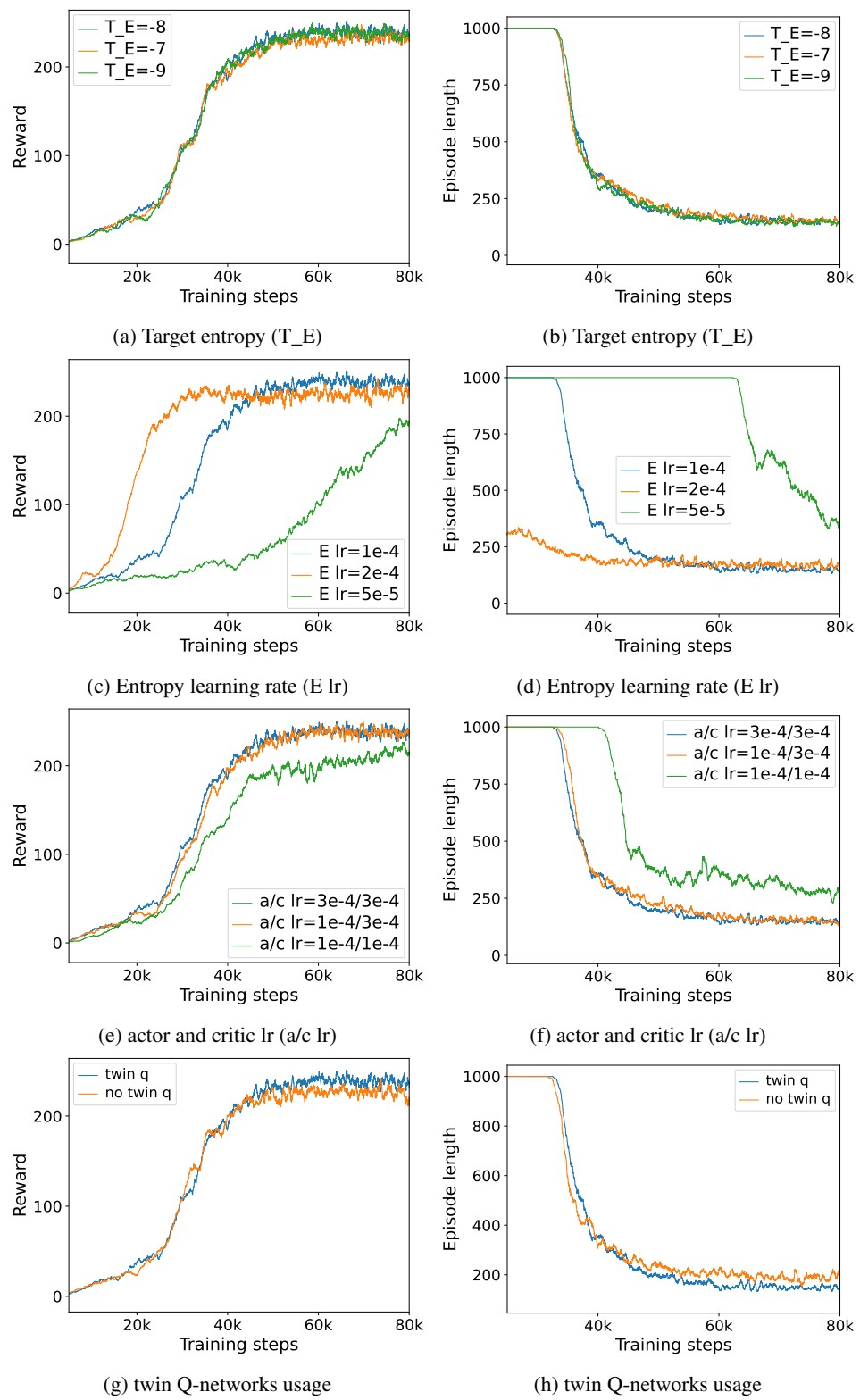

(a) Target entropy (T_E)

(b) Target entropy (T_E)

(c) Entropy learning rate (E lr)

(d) Entropy learning rate (E lr)

(e) actor and critic lr (a/c lr)

(f) actor and critic lr (a/c lr)

(g) twin Q-networks usage

(h) twin Q-networks usage

Figure 11: Performance comparison of MACS with different values of the target entropy, entropy/actor/critic learning rates, and twin Q-networks flag. The blue lines always indicate the values used in this study.

# B  Additional experimental details and results

## B.1  Hardware usage for the training and testing purposes

We train MACS for ~80000 training steps in total using 40 concurrently running environments on a Linux cluster node equipped with two 20-core Intel(R) Xeon(R) Gold 6138 CPUs (2.00 GHz) and 384 GB of memory. All baselines are not data-driven or trained; they are based on explicit, deterministic logic.

Optimization of the test sets using MACS and the baselines was performed on the same hardware used to train MACS. However, only a single CPU core was used for the optimization of the test sets, while the remaining 39 cores remained idle. This setup was chosen to ensure a fair comparison by preventing MACS from gaining an advantage through parallelization, as the baselines are not parallelized.

## B.2  Training time contribution to the experiments

The training time of MACS is 343,800 seconds and was not included in its evaluation. We evaluate the MACS policy based only on optimization time because including the combined training and testing time would make the evaluation highly sensitive to the size and number of test sets. In fact, the experiments could be extended by increasing the sizes/numbers of the test sets until the relative contribution of training time becomes negligible, leading to the reported results.

Moreover, this setup reflects practical usage, as screening hundreds of thousands of structures is a typical use case in computational chemistry. Furthermore, the scalability and zero-shot transferability of MACS allow it to be trained on one set of compositions and then used to efficiently optimize larger structures from other compositions, without additional training, i.e., with zero training time in those cases.

It is ambiguous how to estimate the combined training and testing time, as these phases were conducted under different conditions. Although the same hardware was used, the difference in the number of active cores (see Appendix B.1) raises the question of whether to report runtime using wall-clock time or CPU time, and which approach is fairer.

For reference, comparing the combined training and optimization time of MACS with the optimization time of the fastest baseline (BFGSLS), MACS is 3% faster in terms of wall-clock time but 126 times slower in terms of CPU time. Given the fact that training time can be omitted when the method is used in practice, we compare the methods only by the optimization time, as shown in Tab. 1.

## B.3  Baselines

The baselines are accessed through the CHGNet package, which in turn interfaces with the optimization methods provided by the ASE package. We use the BFGS, BFGSLS, FIRE, and MDMin implemented in ASE, while CG is implemented in SciPy and accessed through the CHGNet $\rightarrow$ ASE chain. The hyperparameters of the baselines are well-tuned and commonly used without modification; we adopt them as-is.

## B.4  Random structures generation

We generate training and testing structures in the following way. Given a composition, a number of atoms, and a parameter $v$, we use the AIRSS package to generate a pseudo-random structure. AIRSS creates a unit cell with a random volume in the range $[v - 5\%, v + 5\%]$ and places the atoms within this unit cell so that the minimum distance between any two atoms is 1Å.

During training, at the beginning of each episode, we randomly select a composition from the list of training compositions. We use the volume of the experimental structure for a given composition as the parameter $v$ in both training and testing to reflect the physics of the real material. Another parameter is the number of atoms in a structure, which is selected to be closest to $40$ atoms, given the composition.

Table 5: The averaged discounted reward after 80,000 training steps.

| Policy | Converged discounted episodic reward |
|---|---|
| Seed 1 (used in the paper) | 236.0 |
| Seed 2 | 237.4 |
| Seed 3 | 235.6 |
| Seed 4 | 234.8 |
| Seed 5 | 235.2 |
| Mean (std) | 235.8 (0.9) |

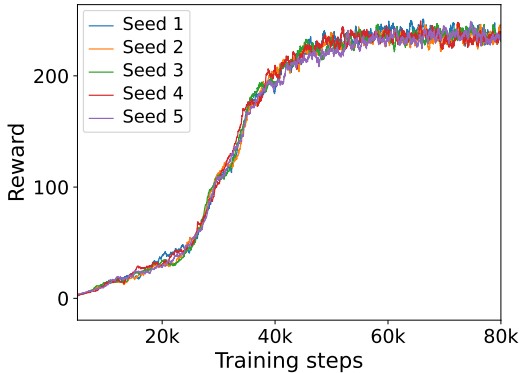

Figure 12: The discounted reward evolution for training MACS starting from five random seeds.

## B.5 Variability of the MACS Policy

To identify the variability of the policy trained by MACS, we independently trained five policies Seed 1 – Seed 5, each starting from a different random seed while using the same hyperparameters and the time for training. Then we compared the converged discounted episodic rewards Tab. 5 and their evolutions in Fig. 12. The table shows that the policy trained by MACS consistently produces similar results across different random seeds.

## B.6 Additional experimental data

Tabs. 6–8 extend the data presented in Tab. 1. Specifically, Tab. 6 and Tab. 7 compare MACS and all baselines in terms of $\mathbf{T}_{mean}$ and $\mathbf{N}_{mean}$, respectively, including standard errors across all test sets, while Tab. 8 reports $\mathbf{P}_F$ for all test sets and methods. Figs. 13 and 14 show the energy distribution for the local minima obtained by the different methods for all test sets. Figs. 15 and 16 show the energy evolution averaged by all successful optimizations of all test sets by the different methods.

## B.7 The analysis of structures optimized by MACS

We calculated the atom-atom distances up to 6.00Å within all 7,200 structures which were optimized using MACS. For each composition, we tabulated the distances, rounded to 2 decimal places as histograms, all of the histograms for individual compositions were then summed to give a total atom-atom distance histogram for each composition. For each composition, we have then tabulated the shortest distance observed, and the distance of the first major peak in the histogram, both provided in Angstroms. We provide these results in Tab. 9.

The shortest interatomic distances observed are plausible distances, the major observed peaks for each composition correspond to expected values when considering each chemistry. For example, the first peak at 1.87Å in $SrTiO_3$, is close to that of the Ti-O distances in the experimentally observed structure of 1.95Å. These observations in addition to the distribution of energies shown in Fig. 13 lead us to the conclusion that the final structures produced by MACS do not contain unphysically short interatomic distances typical for the structures produced due to the failure of the energy calculator.

Table 6: Comparison of MACS and the baselines by $\mathbf{T}_{mean}$ on all test sets. Standard errors are in brackets.

| Composition | N atoms | $\mathbf{T_{mean}}$ (sec) | | | | | | |
| | | MACS | BFGS | FIRE | MDMin | BFGSLS | FIRE+BFGSLS | CG |
|---|---|---|---|---|---|---|---|---|
| $Y_2O_3$ | 40 | **18(1)** | 48(2) | 42(2) | 74(2) | 28(2) | 38(2) | 64(4) |
| | 60 | **32(1)** | 85(3) | 92(3) | 155(3) | 70(7) | 74(2) | 118(6) |
| | 80 | **48(1)** | 137(4) | 130(4) | 207(4) | 97(7) | 112(4) | 184(9) |
| $Cu_{28}S_{16}$ | 44 | **29(1)** | 62(2) | 47(1) | 116(3) | 54(12) | 47(1) | 81(3) |
| | 66 | **51(1)** | 112(3) | 88(2) | 205(4) | 60(1) | 79(2) | 147(5) |
| | 88 | **74(2)** | 175(4) | 120(2) | 315(4) | 110(2) | 117(2) | 275(7) |
| $SrTiO_3$ | 40 | **57(12)** | 94(3) | 109(3) | 248(5) | 65(5) | 102(1) | 134(7) |
| | 60 | **90(12)** | 163(6) | 202(5) | 461(6) | 138(10) | 200(21) | 250(13) |
| | 80 | **142(13)** | 242(8) | 366(8) | 672(7) | 214(16) | 332(17) | 452(20) |
| $Ca_3Ti_2O_7$ | 48 | **59(12)** | 135(5) | 119(3) | 264(5) | 68(3) | 120(4) | 156(7) |
| | 72 | **106(12)** | 199(6) | 239(4) | 479(7) | 184(18) | 252(5) | 301(14) |
| | 96 | **163(12)** | 276(7) | 412(14) | 705(7) | 195(6) | 324(13) | 499(21) |
| $K_3Fe_5F_{15}$ | 46 | **31(1)** | 82(3) | 124(5) | 155(3) | 68(5) | 98(6) | 132(7) |
| | 69 | **51(1)** | 146(4) | 214(8) | 276(5) | 118(19) | 181(16) | 248(11) |
| | 92 | **96(12)** | 236(6) | 377(12) | 485(8) | 154(9) | 271(17) | 371(16) |
| $Ca_3Al_2Si_3O_{12}$ | 40 | **117(4)** | 127(5) | 226(7) | 585(5) | 167(23) | 190(5) | 307(17) |
| | 60 | **237(14)** | 266(8) | 389(14) | 1068(7) | 276(16) | 422(13) | 572(31) |
| | 80 | 343(16) | **333(8)** | 627(18) | 1279(6) | 400(7) | 627(17) | 958(47) |
| **Compositions unseen during the training** | | | | | | | | |
| $Sr_2TiO_4$ | 56 | **65(2)** | 145(5) | 174(5) | 361(5) | 105(10) | 139(2) | 234(12) |
| | 112 | **189(13)** | 358(10) | 665(18) | 759(6) | 414(63) | 440(8) | 559(23) |
| $Sr_3Ti_2O_7$ | 48 | **54(1)** | 123(4) | 151(4) | 315(5) | 88(6) | 130(1) | 191(9) |
| | 96 | **159(12)** | 369(10) | 497(10) | 800(6) | 366(65) | 375(6) | 477(20) |
| $Sr_4Ti_3O_{10}$ | 34 | **30(1)** | 77(3) | 87(3) | 174(4) | 48(3) | 76(3) | 106(6) |
| | 68 | **113(12)** | 202(6) | 238(6) | 498(7) | 149(23) | 205(4) | 307(16) |

## B.8 Composition targeted training

We train MACS on the only composition ($Ca_3Al_2Si_3O_{12}$) in which MACS was outperformed by a baseline (BFGS). We train the policy for the same period of time as we did for the cross-composition training. Tab. 10 confirms that MACS trained specifically on $Ca_3Al_2Si_3O_{12}$ structures is superior to BFGS and MACS trained across all compositions in the training set in all metrics. The composition $Ca_3Al_2Si_3O_{12}$ has the largest number of species, and its observation space can be more diverse than those of other compositions in this study. We suggest that longer training or increasing the proportions of complex compositions during training can help MACS optimize them better.

## B.9 Additional ablation study

**Observations.** MACS and feat.9 achieve similar mean episodic reward and mean episode length during training, as well as similar performance in optimizing $SrTiO_3$ structures. We proceed with the optimization of all test sets using the feat.9 design and compare the results of MACS and feat.9 in Tab. 11. We can see that the two policies are comparable in terms of $\mathbf{T}_{mean}$, while MACS achieves an $\mathbf{N}_{mean}$ that is 7.7% lower than that of feat.9, confirming the superiority of MACS. [*]

---

[*]The optimization of the test set with the $Cu_{28}S_{16}$ structures of 88 atoms was interrupted by the cluster maintenance works and could not be finished due to the replacement of the hardware used for testing. Although optimization of the remaining structures of this test set could change the aggregated numbers, we rely on the comparison between all test sets to conclude that MACS consistently outperforms feat.9 as it consistently requires a lower number of energy calculations than feat.9.

Table 7: Comparison of MACS and the baselines by $N_{mean}$ and $C_{mean}$ on all test sets. The metrics $N_{mean}$ and $C_{mean}$ are presented either as a single value (if they are identical) or separated by a ';' symbol (if they differ). Standard errors are in the brackets.

| Composition | N atoms | $N_{mean}$ ; $C_{mean}$ | | | | | | |
|---|---|---|---|---|---|---|---|---|
| | | MACS | BFGS | FIRE | MDMin | BFGSLS | FIRE+ BFGSLS | CG |
| $Y_2O_3$ | 40 | **121(3)** | 313(10) | 262(4) | 442(11) | 137(2) ; 185(7) | 252(3) ; 267(6) | 122(2) ; 543(10) |
| | 60 | **147(3)** | 340(10) | 338(5) | 553(12) | 178(5) ; 281(23) | 324(4) ; 357(9) | 145(3) ; 642(12) |
| | 80 | **169(3)** | 395(10) | 393(4) | 625(14) | 206(5) ; 307(20) | 360(5) ; 403(14) | 171(3) ; 754(15) |
| $Cu_{28}S_{16}$ | 44 | **150(3)** | 293(6) | 257(4) | 543(11) | 147(3) ; 177(6) | 232(3) ; 242(3) | 158(3) ; 716(13) |
| | 66 | **186(4)** | 355(8) | 307(5) | 633(11) | 176(3) ; 201(4) | 280(3) ; 291(4) | 198(4) ; 881(20) |
| | 88 | **230(5)** | 414(7) | 392(6) | 745(12) | 213(4) ; 239(4) | 352(3) ; 365(4) | 283(6) ; 1269(26) |
| $SrTiO_3$ | 40 | **143(3)** | 255(8) | 314(5) | 625(12) | 133(3) ; 190(16) | 276(2) ; 299(3) | 141(3) ; 572(13) |
| | 60 | **179(4)** | 316(10) | 379(6) | 719(14) | 169(3) ; 255(14) | 321(3) ; 406(48) | 168(4) ; 681(16) |
| | 80 | **208(5)** | 329(8) | 446(7) | 765(17) | 199(4) ; 317(26) | 355(3) ; 433(24) | 205(5) ; 837(21) |
| $Ca_3Ti_2O_7$ | 48 | **146(3)** | 270(9) | 324(5) | 623(13) | 136(2) ; 185(10) | 284(2) ; 317(11) | 151(3) ; 618(12) |
| | 72 | **183(4)** | 310(8) | 408(6) | 707(15) | 168(3) ; 249(27) | 335(3) ; 374(7) | 186(3) ; 756(12) |
| | 96 | **205(4)** | 353(9) | 467(6) | 762(18) | 193(3) ; 267(9) | 369(3) ; 447(23) | 213(4) ; 876(15) |
| $K_3Fe_5F_{15}$ | 46 | **111(2)** | 274(8) | 246(3) | 501(10) | 135(3) ; 178(7) | 246(4) ; 263(6) | 134(3) ; 602(16) |
| | 69 | **128(2)** | 320(7) | 293(4) | 596(11) | 163(4) ; 259(53) | 299(5) ; 344(28) | 160(3) ; 720(12) |
| | 92 | **143(2)** | 359(6) | 326(4) | 642(10) | 176(4) ; 236(13) | 353(5) ; 396(16) | 181(4) ; 815(20) |
| $Ca_3Al_2Si_3O_{12}$ | 40 | 209(5) | **189(5)** | 307(6) | 700(31) | 141(4) ; 264(46) | 269(3) ; 311(6) | 147(3) ; 553(15) |
| | 60 | 264(7) | **230(5)** | 382(6) | 755(55) | 165(3) ; 296(16) | 316(3) ; 391(14) | 177(4) ; 669(15) |
| | 80 | 317(8) | **246(6)** | 461(7) | 894(0) | 189(3) ; 327(6) | 350(3) ; 458(16) | 214(6) ; 814(23) |
| **Compositions unseen during the training** | | | | | | | | |
| $Sr_2TiO_4$ | 56 | **172(4)** | 335(10) | 371(6) | 700(14) | 162(3) ; 218(7) | 319(3) ; 353(6) | 175(4) ; 716(15) |
| | 112 | **245(5)** | 420(10) | 554(8) | 850(16) | 242(5) ; 397(35) | 427(4) ; 508(10) | 245(4) ; 1003(19) |
| $Sr_3Ti_2O_7$ | 48 | **153(4)** | 288(8) | 345(5) | 676(12) | 153(3) ; 210(10) | 299(2) ; 323(3) | 167(4) ; 682(17) |
| | 96 | **227(5)** | 382(9) | 501(7) | 817(20) | 212(4) ; 343(31) | 385(3) ; 449(7) | 223(5) ; 909(19) |
| $Sr_4Ti_3O_{10}$ | 34 | **126(3)** | 251(8) | 282(5) | 547(13) | 124(3) ; 173(14) | 256(3) ; 275(3) | 137(4) ; 557(17) |
| | 68 | **186(5)** | 310(8) | 408(6) | 729(14) | 179(4) ; 304(62) | 339(3) ; 385(8) | 183(3) ; 743(14) |

Table 8: Comparison of MACS and the baselines by $P_F$ (%).

| Composition | N atoms | MARL | BFGS | FIRE | MDMin | BFGSLS | FIRE+ BFGSLS | CG |
|---|---|---|---|---|---|---|---|---|
| $Y_2O_3$ | 40 | **0.33** | 5.67 | 7 | 17.33 | **0.33** | 1.33 | 19 |
| | 60 | **0** | 4.33 | 16 | 28.67 | 1 | 0.67 | 26.67 |
| | 80 | 0.67 | 12.67 | 18.33 | 39.33 | **0.33** | 1.33 | 32.33 |
| $Cu_{28}S_{16}$ | 44 | 0.33 | 3.67 | 1 | 4 | 0.67 | **0** | 6.67 |
| | 66 | 0.33 | 2.67 | 2.67 | 15 | **0** | **0** | 11 |
| | 88 | **0** | **0** | 0.33 | 56.67 | **0** | **0** | 3.33 |
| $SrTiO_3$ | 40 | **0** | 2.33 | 3.33 | 25 | **0** | 0.33 | 15.67 |
| | 60 | **0.33** | 2.33 | 5 | 46.33 | 0.67 | 0.67 | 21 |
| | 80 | **0.33** | 3 | 9.33 | 71.67 | **0.33** | 1.33 | 26.67 |
| $Ca_3Ti_2O_7$ | 48 | **0** | 3.67 | 1.33 | 22.33 | **0** | **0** | 10.33 |
| | 72 | **0** | 0.67 | 1.67 | 54 | 0.33 | 0.33 | 13 |
| | 96 | 0.33 | 0.67 | 3.33 | 72.33 | **0** | **0** | 20 |
| $K_3Fe_5F_{15}$ | 46 | **0** | 1.33 | 17.67 | 4.67 | 0.33 | 4.33 | 14 |
| | 69 | **0** | 1.33 | 22.67 | 14 | **0** | 3 | 19.33 |
| | 92 | **0** | 2.33 | 33.67 | 20.33 | 0.33 | 4 | 20 |
| $Ca_3Al_2Si_3O_{12}$ | 40 | 1 | 1.33 | 5 | 90 | **0** | 1 | 19 |
| | 60 | 3 | 1.33 | 3 | 97 | 0.33 | **0** | 24.33 |
| | 80 | 1.33 | **0** | 4.67 | 99.67 | **0** | 0.33 | 27 |
| **Compositions unseen during the training** | | | | | | | | |
| $Sr_2TiO_4$ | 56 | 0.33 | 3.33 | 14 | 46.33 | 1.33 | **0** | 17.67 |
| | 112 | **0** | 5.67 | 31 | 90 | 1.33 | **0** | 20.67 |
| $Sr_3Ti_2O_7$ | 48 | **0** | 2.67 | 4 | 38.33 | 0.33 | 0.33 | 17.67 |
| | 96 | **0** | 3.67 | 14 | 81.33 | 1 | **0** | 20.33 |
| $Sr_4Ti_3O_{10}$ | 34 | 0.33 | 5 | 4 | 18.67 | **0** | 0.67 | 14.67 |
| | 68 | **0** | 2.33 | 8.33 | 55.67 | **0** | **0** | 17 |

Table 9: The shortest atom-atom distance observed and the distance of the first major peak in Å for the structures optimized by MACS. The total number of counts for the distances observed in brackets.

| Composition | Min distance | First peak |
|---|---|---|
| $Ca_3Al_2Si_3O_{12}$ | 1.44 (4) | 1.69 (5324) |
| $Ca_3Ti_2O_7$ | 1.38 (2) | 1.85 (3924) |
| $Cu_{28}S_{16}$ | 1.97 (2) | 2.22 (4708) |
| $K_3Fe_5F_{15}$ | 1.78 (2) | 2.00 (5392) |
| $Y_2O_3$ | 1.45 (2) | 2.22 (8218) |
| $SrTiO_3$ | 1.40 (4) | 1.87 (3898) |
| $Sr_2TiO_4$ | 1.41 (12) | 1.84 (2850) |
| $Sr_3Ti_2O_7$ | 1.41 (8) | 1.86 (2506) |
| $Sr_4Ti_3O_{10}$ | 1.41 (8) | 1.86 (1950) |

Table 10: Comparison of the MACS policy trained on the $Ca_3Al_2Si_3O_{12}$ composition (MACS individual) with the MACS policy trained across all compositions in the training set (MACS) and BFGS.

| Composition | N atoms | $T_{mean}$ (sec) | | | $N_{mean}$ | | |
|---|---|---|---|---|---|---|---|
| | | MACS | MACS individual | BFGS | MACS | MACS individual | BFGS |
| $Ca_3Al_2Si_3O_{12}$ | 40 | 117 | **85** | 127 | 209 | **153** | 189 |
| | 60 | 237 | **170** | 266 | 264 | **205** | 230 |
| | 80 | 343 | **237** | 333 | 317 | **229** | 246 |
| AVERAGE | 60 | 232 | **164** | 242 | 264 | **196** | 222 |
| $P_F$ (%) | | 1.78 | **0.33** | 0.89 | | | |

Table 11: Comparison of the feat.9 design with MACS by $T_{mean}$ and $N_{mean}$. Standard errors are in brackets.

| Composition | N atoms | $T_{mean}$ (sec) | | $N_{mean}$ | |
|---|---|---|---|---|---|
| | | MACS | feat.9 | MACS | feat.9 |
| $Y_2O_3$ | 40 | **18(1)** | 22(1) | **121(3)** | 143(5) |
| | 60 | **32(1)** | 39(1) | **147(3)** | 168(4) |
| | 80 | **48(1)** | 61(2) | **169(3)** | 211(6) |
| $Cu_{28}S_{16}$ | 44 | **29(1)** | **29(1)** | **150(3)** | 158(4) |
| | 66 | **51(1)** | 54(2) | **186(4)** | 214(6) |
| | 88 | 74(2) | **67(3)** | **230(5)** | 338(30) |
| $SrTiO_3$ | 40 | 57(12) | **44(1)** | **143(3)** | 145(3) |
| | 60 | 90(12) | **81(2)** | **179(4)** | 184(5) |
| | 80 | 142(13) | **132(3)** | **208(5)** | 209(4) |
| $Ca_3Ti_2O_7$ | 48 | 59(12) | **56(1)** | **146(3)** | 154(3) |
| | 72 | 106(12) | **98(2)** | **183(4)** | **183(4)** |
| | 96 | 163(12) | **134(3)** | **205(4)** | 214(4) |
| $K_3Fe_5F_{15}$ | 46 | **31(1)** | 38(1) | **111(2)** | 141(3) |
| | 69 | **51(1)** | 64(2) | **128(2)** | 167(4) |
| | 92 | **96(12)** | 112(2) | **143(2)** | 183(4) |
| $Ca_3Al_2Si_3O_{12}$ | 40 | 117(4) | **111(3)** | 209(5) | **189(5)** |
| | 60 | 237(14) | **195(5)** | 264(7) | **245(7)** |
| | 80 | 343(16) | **334(8)** | 317(8) | **292(7)** |
| **Compositions unseen during training** | | | | | |
| $Sr_2TiO_4$ | 56 | 65(2) | **63(1)** | **172(4)** | **172(4)** |
| | 112 | 189(13) | **174(4)** | **245(5)** | 247(5) |
| $Sr_3Ti_2O_7$ | 48 | 54(1) | **53(1)** | **153(4)** | 155(4) |
| | 96 | **159(12)** | **159(4)** | **227(5)** | 229(5) |
| $Sr_4Ti_3O_{10}$ | 34 | **30(1)** | 31(1) | **126(3)** | 131(3) |
| | 68 | 113(12) | **88(2)** | **186(5)** | 190(4) |
| AVERAGE | 66 | **100** | 101 | **181** | 195 |

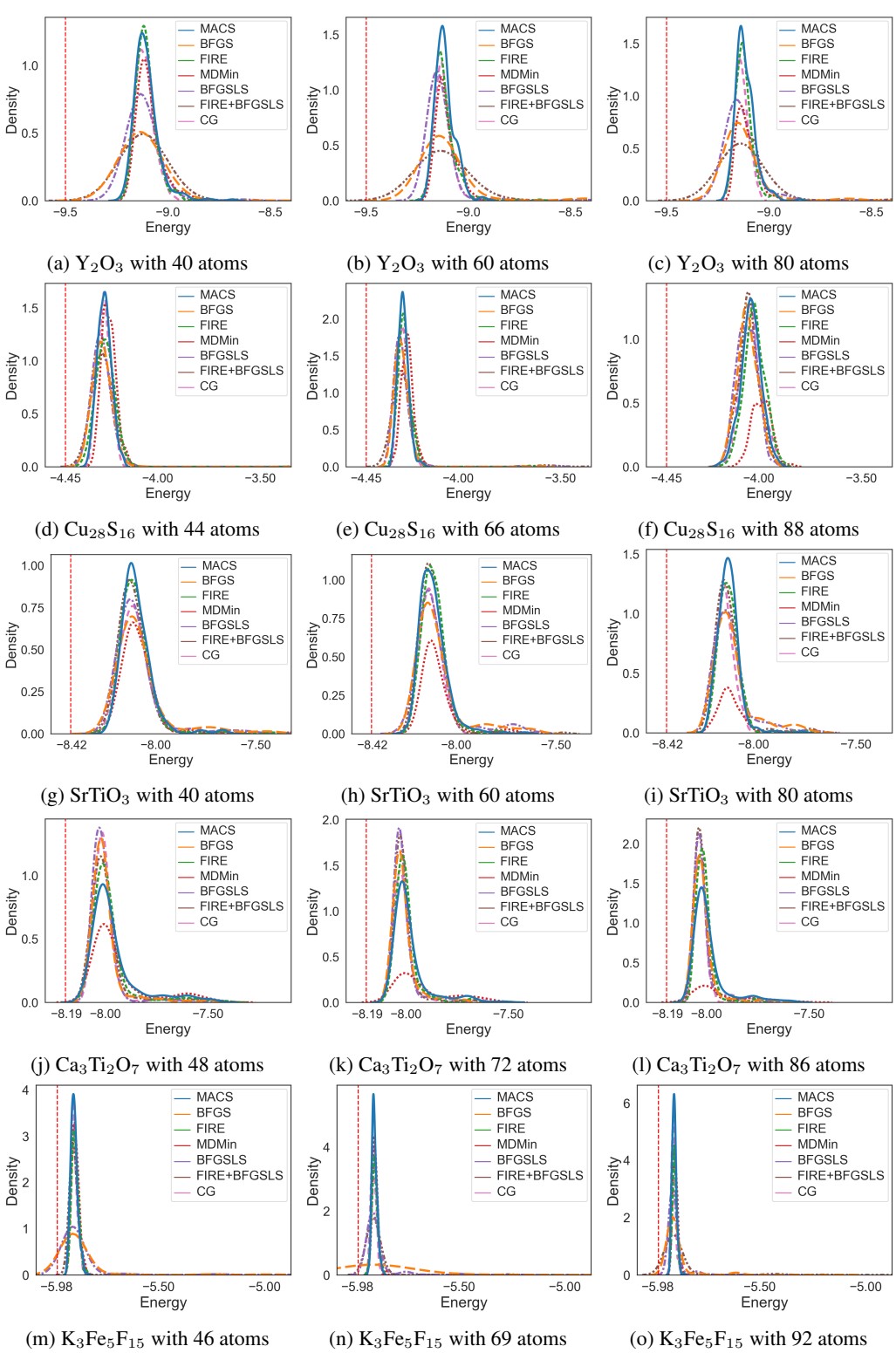

Figure 13: The energy distribution of the local minima obtained by all different methods. The vertical line indicates the energy of the experimental structure.

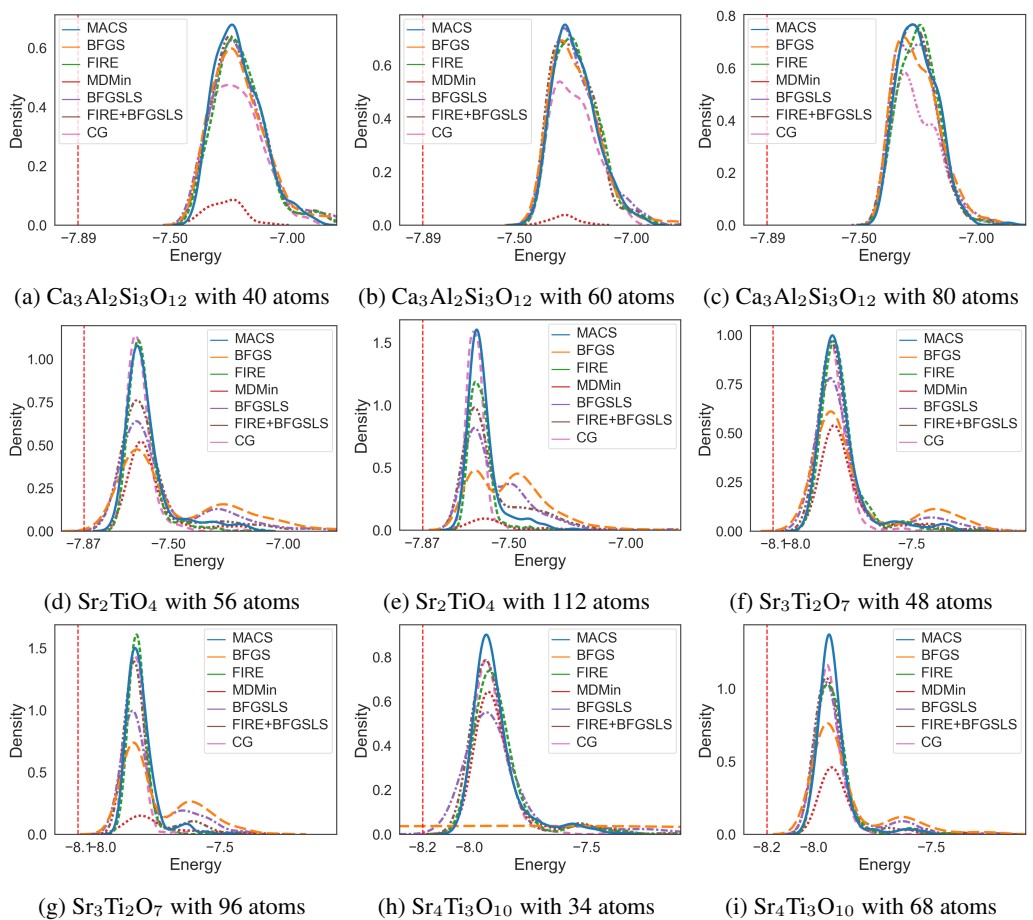

(a) Ca$_3$Al$_2$Si$_3$O$_{12}$ with 40 atoms    (b) Ca$_3$Al$_2$Si$_3$O$_{12}$ with 60 atoms    (c) Ca$_3$Al$_2$Si$_3$O$_{12}$ with 80 atoms

(d) Sr$_2$TiO$_4$ with 56 atoms    (e) Sr$_2$TiO$_4$ with 112 atoms    (f) Sr$_3$Ti$_2$O$_7$ with 48 atoms

(g) Sr$_3$Ti$_2$O$_7$ with 96 atoms    (h) Sr$_4$Ti$_3$O$_{10}$ with 34 atoms    (i) Sr$_4$Ti$_3$O$_{10}$ with 68 atoms

Figure 14: The energy distribution of the local minima obtained by all different methods. The vertical line indicates the energy of the experimental structure.

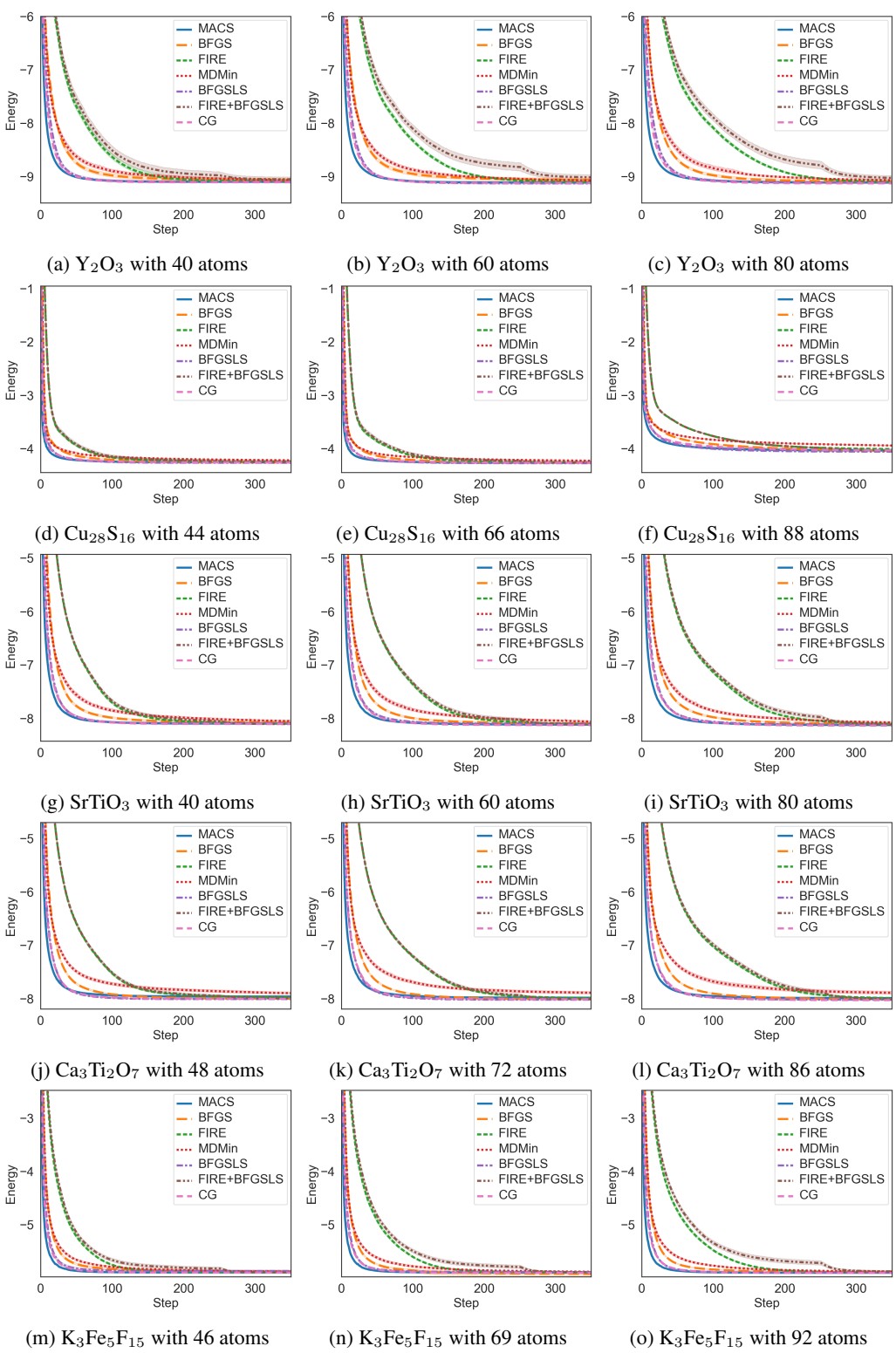

Figure 15: Energy evolution averaged over all successfully optimized structures for all methods on the given test set.

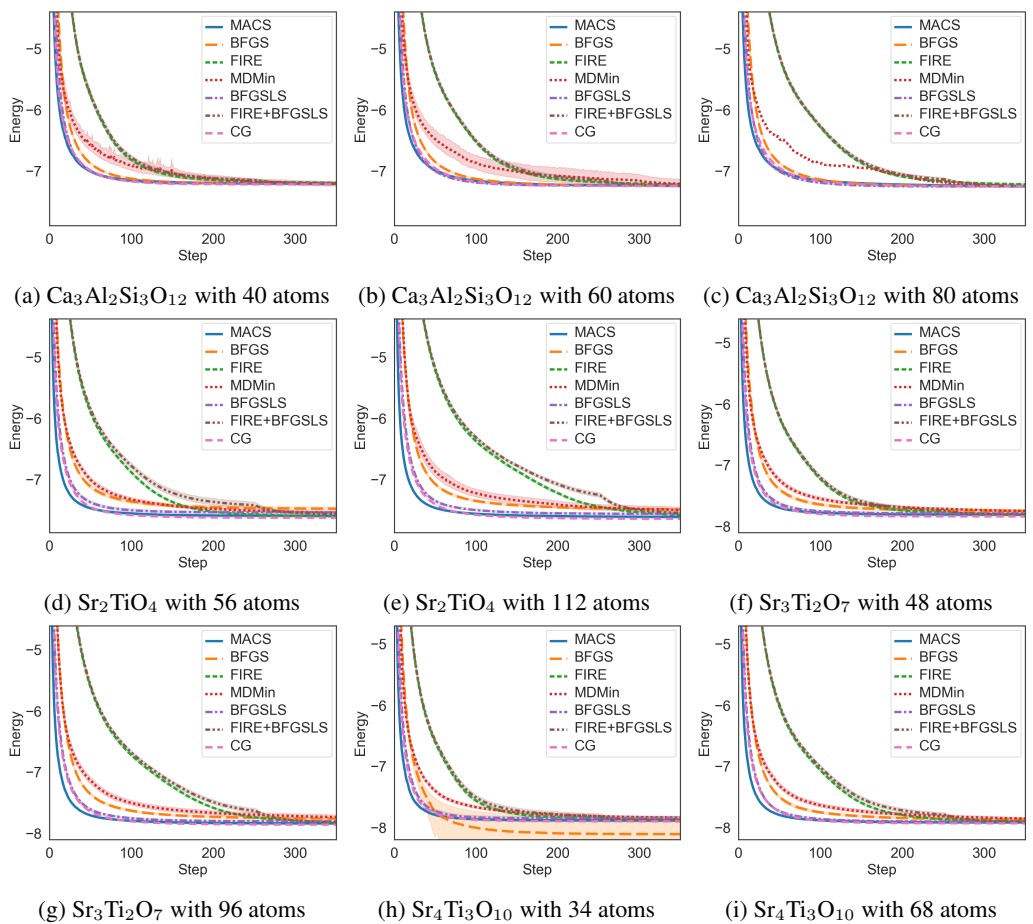

(a) $Ca_3Al_2Si_3O_{12}$ with 40 atoms     (b) $Ca_3Al_2Si_3O_{12}$ with 60 atoms     (c) $Ca_3Al_2Si_3O_{12}$ with 80 atoms

(d) $Sr_2TiO_4$ with 56 atoms     (e) $Sr_2TiO_4$ with 112 atoms     (f) $Sr_3Ti_2O_7$ with 48 atoms

(g) $Sr_3Ti_2O_7$ with 96 atoms     (h) $Sr_4Ti_3O_{10}$ with 34 atoms     (i) $Sr_4Ti_3O_{10}$ with 68 atoms

Figure 16: Energy evolution averaged over all successfully optimized structures for all methods on the given test set.

