# OpenReview forum: "MACS: Multi-Agent Reinforcement Learning for Optimization of Crystal Structures"
_NeurIPS.cc/2025/Conference — NeurIPS 2025 poster_

### Official Review · Reviewer_TGFN · 2025-06-22

**Clarity:** 2
**Significance:** 1
**Originality:** 2
**Rating:** 3
**Confidence:** 2

**Summary:**

This paper presents MACS, a novel multi-agent reinforcement learning (MARL) method for optimizing periodic crystal structures. By formulating the geometry optimization task as a partially observable Markov game (POMG), the method treats atoms as decentralized agents and learns local policies that collectively minimize atomic forces in periodic structures. The authors benchmark MACS against state-of-the-art optimization methods such as BFGS, FIRE, CG, and others, and demonstrate significant improvements in optimization time, failure rate, and transferability to unseen systems.

**Questions:**

N.A.

**Ethical Concerns:**

["NO or VERY MINOR ethics concerns only"]

**Final Justification:**

Part of the unclearness is explained, but most of the weaknesses are still unaddressed.

**Limitations:**

N.A.

**Paper Formatting Concerns:**

N.A.

**Quality:**

2

**Strengths And Weaknesses:**

Strength:

1. The problem of crystal structure optimization is well-motivated, and the experimental setup includes a reasonable benchmark against well-known baselines.

2. Experimental results suggest improved generalization performance over several baselines.

Weakness:

1. Although the paper casts atoms as agents in a MARL setting, the agent design lacks meaningful interaction or interdependence typically expected in multi-agent systems. There is no explicit coordination mechanism beyond standard parameter sharing. This setup resembles multiple parallel optimizers more than a principled MARL approach.

2. The “agent” in MACS functions more like a local function approximator on force minimization than a true autonomous or strategic decision-maker in a multi-agent setting. The use of the MARL framework here feels superficial and may be misleading in terms of the novelty claimed.

3. The paper compares against standard MARL baselines, but does not include strong compositional or hierarchical RL baselines

4. No theoretical insights or guarantees are provided on the benefits of compositionality, generalization bounds, or conditions under which MACS would outperform flat policies.

---

> ### Author Rebuttal · Authors · 2025-07-30
>
> Dear Reviewer,
>
> Thank you for taking the time to read our paper and provide valuable feedback.
>
> > Weakness: Although the paper casts atoms as agents in a MARL setting, the agent design lacks meaningful interaction or interdependence typically expected in multi-agent systems. There is no explicit coordination mechanism beyond standard parameter sharing. This setup resembles multiple parallel optimizers more than a principled MARL approach.
>
> > Weakness: The “agent” in MACS functions more like a local function approximator on force minimization than a true autonomous or strategic decision-maker in a multi-agent setting. The use of the MARL framework here feels superficial and may be misleading in terms of the novelty claimed.
>
> **Answer to Weakness 1**. We would like to refer the reviewer to our Summary Rebuttal (at the beginning of the Rebuttal to ytGL) that addresses these questions.
>
> > Weakness: The paper compares against standard MARL baselines, but does not include strong compositional or hierarchical RL baselines
>
> **Answer to Weakness 2**. To the best of our knowledge, we are the first to apply MARL to address periodic crystal structure optimization (as mentioned in Lines 58-59 in our paper). All baselines are not MARL approaches, and we are not aware of any existing hierarchical RL methods used for this task. We believe hierarchical RL methods could be more suitable for solving global structure optimization, which is an interesting direction for future work.
>
> > Weakness: No theoretical insights or guarantees are provided on the benefits of compositionality, generalization bounds, or conditions under which MACS would outperform flat policies.
>
> **Answer to Weakness 3**.  We appreciate the reviewer’s point and agree that providing formal theoretical guarantees for performance or generalization in this setting remains an open challenge. Our primary contribution focuses on designing and validating an MARL-based framework that addresses the complexity of crystal structure optimization, which involves highly non-convex, high-dimensional energy landscapes with strong physical constraints. Deriving theoretical guarantees in such environments is non-trivial because reward functions and state spaces depend on material-specific physics rather than the synthetic or formal assumptions typically used in theoretical analysis.
>
> That said, our results provide strong empirical evidence that the MARL framework preserves transferability within a phase field and maintains scalability for structures twice as large as the structures the policy was trained on. We agree that establishing theoretical conditions for guarantees on optimization time and quality - whether for existing methods or new ones - would be a significant achievement, which we leave for future research.

---

> > ### Comment · Reviewer_TGFN · 2025-08-05
> >
> > Thanks for the rebuttal! Part of my concerns have been addressed, such as the weakness 2. The other weaknesses are still there or not fully addressed. I will increase my score to 3 based on the current revision.

---

> > > ### Author Response · Authors · 2025-08-06
> > >
> > > Thank you for your comments, and for raising your score. If possible, could you please state what your remaining concerns are so that we can try to address them?

---

### Official Review · Reviewer_V57T · 2025-06-26

**Clarity:** 3
**Significance:** 2
**Originality:** 2
**Rating:** 4
**Confidence:** 3

**Summary:**

This paper introduces a multi-agent RL approach to optimizing crystal structure. The authors propose using independent SAC agents to represent each atom in the structure and introduce a state space which provides local information to each agent. Each agent is trained on independant reward functions simultaneously. The authors measure the performance of their proposed method against existing optimization techniques and find that their proposed method can achieve as good results as the best baseline while being significantly more efficient.

**Questions:**

(1) I am not 100% clear if the training computation is included in the metrics for comparing to baselines. My understanding is that it is not. Is this a fair comparison? It would be helpful if the authors could provide a argument for why this makes sense.
(2) Failure rate $P_f$ is not defined and its positioning in Table 1 is a bit confusing. Is this the overall failure rate across all composition? Can this be reported per composition instead?
(3) SAC produces a stochastic policy, however I do not see any discussion of evaulation over multiple seeds - was this done?
(4) In line 295 it states that episodic reward is normalized to compare across reward functions - normalized in what way?

**Ethical Concerns:**

["NO or VERY MINOR ethics concerns only"]

**Final Justification:**

The results across seeds look consistent and my concerns over training time exclusion in the evaluation have been addressed, therefore I will raise my score to 4. I do not raise my score higher as I still think the novelty of the work is somewhat limited, however I think the contribution is technically solid and the performance results are good.

**Limitations:**

The authors address several limitiations, however I would add that the claim in line 319 seems too strong for the results provided in the paper since the test cases only consist of structures that contain the same element compositions as the training cases.

**Quality:**

2

**Strengths And Weaknesses:**

Strengths:
(1) The paper is clearly written and easy to follow. Figure 1 provides a good and clear overview of the algorithm. Sufficient context is provided around crystal structure optimization for someone unfamiliar with the field to understand the application.

(2) Experimental results indicate that the proposed method MACS outperforms existing methods in terms of efficiency while matching the best existing method in terms of failure rate.

Weaknesses:
(1) The novelty of the paper is limited. The primary contribution of the paper is formulating the optimization of crystal structure as a POMDP by defining an appropriate state and action space and reward function. Otherwise, the algorithm is a standard application of SAC. Additionally, while the paper refers to multi-agent RL, the actual algorithm treats each agent as a independent agent optimizing indepedent rewards. While an important and interesting application of RL, the core machine learning contribution is limited.
(2) It is not clear to me if the baseline comparisons are entirely fair. For example - are the reported metrics $T_{mean}$ and $C_{mean}$ inclusive of the training steps required to train MACS? Finally MACS would produce stochastic policies but I do not see a discussion of the variability in policy performance across seeds.

---

> ### Author Rebuttal · Authors · 2025-07-30
>
> Dear Reviewer,
>
> Thank you for all of your kind suggestions and questions. In the following, we answer all of your questions in detail.
>
> > Weakness (1): The novelty of the paper is limited. The primary contribution of the paper is formulating the optimization of crystal structure as a POMDP by defining an appropriate state and action space and reward function. Otherwise, the algorithm is a standard application of SAC. Additionally, while the paper refers to multi-agent RL, the actual algorithm treats each agent as a independent agent optimizing indepedent rewards. While an important and interesting application of RL, the core machine learning contribution is limited.
>
> **Answer to weakness (1)**: We would like to refer the reviewer to the Summary Rebuttal (at the beginning of the Rebuttal to ytGL) for the broad discussion. We acknowledge that exploring richer multi-agent interactions (e.g., via communication or centralized training) is a possible direction for future work.
>
> > Weakness 2: It is not clear to me if the baseline comparisons are entirely fair. For example - are the reported metrics and inclusive of the training steps required to train MACS? Finally MACS would produce stochastic policies but I do not see a discussion of the variability in policy performance across seeds.
>
> > Question (1): I am not 100\% clear if the training computation is included in the metrics for comparing to baselines. My understanding is that it is not. Is this a fair comparison?..
>
> **Answer to Question (1)**: Indeed, the training time is not included in the comparison with the baselines. We suggest explicitly mentioning it in the main text and referring to a new section in the Appendix for the detailed discussion. The proposed text for that section is as follows:
>
> "Appendix Section: Training Time Contribution to the Experiments.
>
> The training time of MACS is 343,800 seconds and was not included in the evaluation of MACS. We evaluate the MACS policy based on optimization time only because including the combined training and testing time would make the evaluation highly sensitive to the size and number of test sets. In fact, the experiments could be extended by increasing the sizes/numbers of the test sets until the relative contribution of training time becomes negligible, leading to the reported results.
>
> Moreover, this setup reflects practical usage, as screening hundreds of thousands of structures is a typical use case in computational chemistry. Furthermore, the scalability and zero-shot transferability of MACS allow it to be trained on one set of compositions and then used to efficiently optimize larger structures from other compositions, without additional training, i.e., with zero training time in those cases.
>
> There is also no fair way to estimate the combined training and testing time, since these phases were conducted under different conditions. Although the same hardware was used the difference in the number of active cores (see Appendix B.1) raises the question of whether to report runtime using wall-clock time or CPU time, and which approach is fairer.
>
> For reference, comparing the combined training and optimization time of MACS with the optimization time of the fastest baseline (BFGSLS), MACS is 3\% faster in terms of wall-clock time but 126 times slower in terms of CPU time. Given the fact that training time can be omitted when the method is used in practice, we compare the methods only by the optimization time, as shown in Table 1."
>
> > Question (2): Failure rate is not defined and its positioning in Table 1 is a bit confusing. Is this the overall failure rate across all composition? Can this be reported per composition instead?
>
> **Answer to Question (2)**. The failure rate notation $\mathbf{P}_{F}$ is introduced on line 240. The failure rate reported in Table 1 represents the average across all test sets. Since the average failure rate is low and comparable to the best baseline, we provide failure rates per test set in Table 7 of Appendix B.4 and refer to it in the caption of Table 1. Table 7 shows that the failure rate remains consistently low across all test sets. We will add a note to Table 1 to clarify that the numbers provided there are averaged across the test sets to avoid the confusion.
>
> > (3) SAC produces a stochastic policy, however I do not see any discussion of evaulation over multiple seeds - was this done?
>
> **Answer to Question (3)**: We initially did not evaluate MACS on the same structures across multiple seeds, as we relied on a large number of randomly generated structures (300 per test set, across 24 test sets) to ensure reliable policy evaluation. Following the reviewer’s suggestion, we have conducted an additional experiment to assess variability due to the stochastic nature of SAC used by MACS. We propose adding the following section to the Appendix and referencing it in the main text:
>
> "Appendix Section: Variability of the MACS Policy.
> We evaluated the policy on a test set containing SrTiO$_3$ structures with 40 atoms by performing nine additional optimization runs, resulting in ten runs in total for this set. We measured variability in the average number of steps for successful optimizations, and the failure rate.
>
> The failure rate was 0.33\% in 2 runs, 0.66\% in 1 run, and 0\% in all others, confirming that failures remain rare.
>
> For the number of optimization steps/energy calculations, we estimated (mean, std) among 10 seeds as (147.48, 3.18) which is well below the mean number of steps (190) in the BFGSLS, the best baseline in this test set."
>
> > Question (4): In line 295 it states that episodic reward is normalized to compare across reward functions - normalized in what way?
>
> **Answer to Question (4)**. All episodic rewards collected during training are normalized using min-max (unity-based) normalization to fit within the range $[0,1]$. This normalization avoids comparisons based on absolute reward values and instead allows assessment of convergence speed and stability.
>
> > Limitations: The authors address several limitiations, however I would add that the claim in line 319 seems too strong for the results provided in the paper since the test cases only consist of structures that contain the same element compositions as the training cases.
>
> **Answer to Limitations**. For any phase field with at least two elements, the number of possible compositions is infinite. In this context, the ability of MACS to apply a policy trained on one composition to optimize all structures within the same phase field makes it a truly universal optimizer for that field. To enable transfer learning across different phase fields, additional training on compositions from diverse phase fields will be required. We suggest to replace the sentence in line 319 by the following sentence:
> "In conclusion, MACS has the potential to evolve into a universal geometric optimizer for periodic crystal structures."

---

> > ### Comment · Reviewer_V57T · 2025-08-01
> >
> > Thank you for addressing my questions. I have a couple follow ups.
> >
> > Regarding the fairness of including the training computation in the comparison to baselines, I believe this depends significantly on how universal the optimizer is at the end of training. I agree that if "this setup reflects practical usage, as screening hundreds of thousands of structures is a typical use case in computational chemistry. Furthermore, the scalability and zero-shot transferability of MACS allow it to be trained on one set of compositions and then used to efficiently optimize larger structures from other compositions, without additional training, i.e., with zero training time in those cases.". You also stated later on that "To enable transfer learning across different phase fields, additional training on compositions from diverse phase fields will be required. ".  I just want to be sure I understand as I am not very familiar with common practice in this field. For a given phase field, it is typical to run hundreds of thousands of optimizations? If so, I agree that excluding training time is fair.
> >
> > Regarding the seed issue, I am not 100% clear if the new experiments have addressed my concern. Have you trained the policy with multiple seeds? I.e. you should have N policies each trained independently with different seeds and each are evaluated in your M evaluation tasks, so NxM evaluations. Otherwise it is possible that some training runs fail to find a good policy due to stochasticity in the training algorithm.

---

> > > ### Author Response · Authors · 2025-08-03
> > >
> > > Dear reviewer, thank you for the quick feedback!
> > >
> > > > I just want to be sure I understand as I am not very familiar with common practice in this field. For a given phase field, it is typical to run hundreds of thousands of optimizations? If so, I agree that excluding training time is fair.
> > >
> > > Indeed, it is common for the phase field exploration and Crystal Structure Prediction, in general, to include hundreds of thousands of structure optimizations (based on the force fields). For instance, in [6], 244 different compositions in the Li-Sn-S-Cl phase field were explored, 1200 structures in each composition.
> > >
> > > > Regarding the seed issue, I am not 100% clear if the new experiments have addressed my concern. Have you trained the policy with multiple seeds?
> > >
> > > We are sorry, we initially misunderstood Question (3) and evaluated the variability of a single policy’s efficiency across different optimizations for one test set. We did not evaluate the variability during training of the final hyperparameter and feature set presented in the paper, as the policy on the final stages of hyperparameter tuning showed a stable performance. We have started training the policy from the different seeds, and we propose adding a paragraph to the Appendix with the findings, which we will draft on August 4–5, depending on the readiness of the results.
> > >
> > > [6] Vasylenko, A., Gamon, J., Duff, B.B. et al. Element selection for crystalline inorganic solid discovery guided by unsupervised machine learning of experimentally explored chemistry. Nat Commun 12, 5561 (2021).

---

> > > > ### Comment · Reviewer_V57T · 2025-08-04
> > > >
> > > > Thank you for the clarifications.
> > > >
> > > > Please let me know when the additional seed experiments are complete and I will update my score accordingly.

---

> > > > > ### Author Response · Authors · 2025-08-08
> > > > >
> > > > > Dear reviewer,
> > > > >
> > > > > Since we are not allowed to submit updated results with images, we present the converged episodic reward for the policy trained with different random seeds in a table. Each policy is trained independently for 80k steps with a different random seed.
> > > > >
> > > > > Seed   |   Converged discounted episodic reward    |
> > > > > ---|---|
> > > > > MACS (from the paper) | 236.0                |
> > > > > Seed 1         | 237.4                |
> > > > > Seed 2           | 235.6                |
> > > > > Seed 3         | 234.8                |
> > > > > Seed 4                 | 235.2                |
> > > > > Mean (std)        | 235.8 (0.9)                |
> > > > > -------------------------------------------------------------------------
> > > > >
> > > > > The table shows that the policy trained by MACS consistently produces similar results across different random seeds. We will accompany the table with the discounted episodic reward evolution plot in the Appendix to confirm the stability of training as well.

---

> > > > > > ### Comment · Reviewer_V57T · 2025-08-08
> > > > > >
> > > > > > Thank you for performing these additional experiments. The results across seeds look consistent and my concerns over training time exclusion in the evaluation have been addressed, therefore I will raise my score.

---

### Official Review · Reviewer_6w1f · 2025-06-27

**Clarity:** 3
**Significance:** 3
**Originality:** 2
**Rating:** 5
**Confidence:** 2

**Summary:**

This paper proposes MACS, a multi-agent reinforcement learning (MARL) framework for geometry optimization of periodic crystal structures. Each atom is modeled as an individual agent, observing local geometric and force-based features and adjusting its position to reduce atomic forces. The optimization is performed using independent SAC and guided by the CHGNet surrogate model for force estimation. The authors report substantial improvements over classical optimization baselines in terms of speed, energy evaluations, and failure rate, and demonstrate zero-shot transferability to larger or unseen compositions.

**Questions:**

See concerns:
- I believe a short discussion would be helpful to better understand how strongly these issues might affect the proposed Markov game.
- The confidence in MACS' performance would be substantially improved if stronger baselines were to be included.
- Would it be possible to confirm the quality of the final crystal structures using DFT?

**Ethical Concerns:**

["NO or VERY MINOR ethics concerns only"]

**Final Justification:**

This is a copy of my last comment to the authors. To summarize:

- my main concern was related to the (in my view) simplistic baselines. This discussion made me realize that the baseline seem appropriate, and thus properly support the benefit of MACS. It also showed my limited knowledge in the application field (as I come from an RL background).

- my second concern focused on the appropriateness of evaluation. This was somewhat addressed by the authors (knowing that it is unreasonable to compute DFT calculations on all final structures during the rebuttal period).

- my third concern was related to the independent nature of the agents in the MA system. Despite the fact that empirically, independent learners have been shown to perform well on SMAC, this does not mean that the MA target problem does not exist and does not affect the current application.

In light of all of this, I will raise my score from 4 to 5, as overall this work seems a solid advancement in a specific subfield of scientific discovery. I will also lower my confidence score from 3 to 2, as I feel I am missing part of the necessary relevant literature.

**Limitations:**

Yes

**Paper Formatting Concerns:**

.

**Quality:**

3

**Strengths And Weaknesses:**

Strengths
---------
- The formulation of geometry optimization as a MARL problem is intuitive, leveraging locality and structure in atomic interactions.
- The pipeline is conceptually simple yet elegant.
- Experiments are extensive, covering diverse compositions, varying sizes, and ablations on observation, action, and reward designs.

Concerns
---------
- To me, the connection to learning to optimize is somewhat superficial: the RL algorithm is standard, the environment (CHGNet + atomic displacements) is fixed, and there is no higher-level learning over tasks.
- The agents operate independently (using independent SAC). While effective, this ignores coordination issues known to affect MARL systems. I believe a short discussion would be helpful to better understand how strongly these issues might affect the proposed Markov game.
- My main concern is that, although the results look strong, the baselines are limited to classical optimizers (e.g., BFGS, FIRE). There has been a number of works on geometric optimization, e.g., [1-3]. I believe the confidence in MACS' performance would be substantially improved if stronger baselines were to be included.
- The energy estimations are based on CHGNet, which is itself a learned model. While fast, it introduces approximation errors. Through exploration, the RL agents might produce crystals that are o.o.d. of CHGNet's training data. It would be valuable to confirm the quality of the final crystal structures using DFT.


[1] Zuo, Y., Qin, M., Chen, C., Ye, W., Li, X., Luo, J., & Ong, S. P. (2021). Accelerating materials discovery with Bayesian optimization and graph deep learning. Materials Today, 51, 126-135.

[2] Li, C. N., Liang, H. P., Zhang, X., Lin, Z., & Wei, S. H. (2023). Graph deep learning accelerated efficient crystal structure search and feature extraction. npj Computational Materials, 9(1), 176.

[3] Chen, S., & Zhang, S. (2022). A structural optimization algorithm with stochastic forces and stresses. Nature Computational Science, 2(11), 736-744.

---

> ### Author Rebuttal · Authors · 2025-07-30
>
> Dear Reviewer,
>
> Thank you for taking the time to read our paper and provide valuable feedback. In the following, we address your concerns and questions in detail.
>
> >Concern 1: To me, the connection to learning to optimize is somewhat superficial: the RL algorithm is standard, the environment (CHGNet + atomic displacements) is fixed, and there is no higher-level learning over tasks.
>
> **Answer 1**. Indeed, MACS is tailored specifically for geometry optimization and does not support optimization tasks from other areas of science. In the manuscript, we note that "an entire field within materials design, known as crystal structure prediction (CSP), focuses on the computational prediction of stable crystal structures" thus geometry optimization plays a fundamental role in computational materials design. Due to the highly diverse chemistry across different compositions and phase fields, we treat optimization within each composition or phase field as an individual task over which MACS learns to optimize.
>
> >Concern 2: The agents operate independently (using independent SAC). While effective, this ignores coordination issues known to affect MARL systems. I believe a short discussion would be helpful to better understand how strongly these issues might affect the proposed Markov game.
>
> **Answer 2**. Summary Rebuttal (at the beginning of the Rebuttal to ytGL) partly addresses this question. We would like to add that previous works [4, 5] have also shown that independent learning methods can perform just as well as or even better than state-of-the-art joint learning MARL approaches on multi-agent benchmark environments like the StarCraft Multi-Agent Challenge (SMAC), which consists of various challenging coordination tasks.
>
> Further, the introduced action scaling coefficient $c_i$, defined by formula (3), restricts the order of magnitude of an agent's action based on its gradient norm. This makes transitions between states more predictable for agents, as each agent can rely on the gradients of neighboring agents in its local observation to estimate how far they **can** move from their current positions. Together with overlapping local observations and interdependent rewards, this enables efficient collective behavior without explicit coordination.
>
> [4] De Witt, C.S., Gupta, T., Makoviichuk, D., Makoviychuk, V., Torr, P.H., Sun, M. and Whiteson, S.. Is independent learning all you need in the starcraft multi-agent challenge? arXiv preprint arXiv:2011.09533, 2020.
>
> [5] Papoudakis, G., Christianos, F., Schäfer, L. and Albrecht, S.V.. Benchmarking multi-agent deep reinforcement learning algorithms in cooperative tasks. In Neural Information Processing Systems Track on Datasets and Benchmarks, 2021.
>
> >Concern 3: My main concern is that, although the results look strong, the baselines are limited to classical optimizers (e.g., BFGS, FIRE). There has been a number of works on geometric optimization, e.g., [1-3]. I believe the confidence in MACS' performance would be substantially improved if stronger baselines were to be included.
>
> > Question 2: The confidence in MACS' performance would be substantially improved if stronger baselines were to be included.
>
> **Answer 3**. We refer the reviewer to the Summary Rebuttal to answer the question. Also, in the papers [1-3], the structure optimization method provided is either designed for a different task or tailored to more restrictive conditions than those addressed in our paper. Specifically, in [1], the optimization method is applied under symmetry constraints to crystal structures **close to local energy minima**; in [2], the authors perform two-step structure optimization using two different energy calculators and **classical optimizers**; in [3], the authors consider the specific problem of structure optimization under **noise in gradient conditions** and introduce a two-staged optimization via the steepest-descent method with a fixed step size on each phase. The steepest descent is known to have slower convergence than other first-order methods, such as CG, one of the baselines in our study.
>
> > Concern 4: The energy estimations are based on CHGNet, which is itself a learned model. While fast, it introduces approximation errors. Through exploration, the RL agents might produce crystals that are o.o.d. of CHGNet's training data. It would be valuable to confirm the quality of the final crystal structures using DFT.
>
> **Answer 4**. The final energy distributions shown in Fig.12 of Appendix B.4 confirm the comparable final energy distribution among results of the optimization by MACS and the baselines. Performing density functional theory calculations on all of the final structures is impractical within the available time scale. However, we do agree with the reviewer, that assessing the quality of the resulting final crystal structures is an important task. For this we have performed an additional analysis of the final structures and suggest to add the following section to the Appendix:
>
> "The analysis of structures optimized by MACS.
>
> We calculated the atom-atom distances up to 6.00 Angstroms within all 7,200 structures which were optimized using MACS. For each composition, we tabulated the distances, rounded to 2 decimal places as histograms, all of the histograms for individual compositions were then summed to give a total atom-atom distance histogram for each composition. For each composition, we have then tabulated the shortest distance observed, and the distance of the first major peak in the histogram, both provided in Angstroms. We provide these results below with the total number of counts for the distances observed in brackets.
>
> Composition   |   Min Distance    |    First peak  |
> ---|---|---|
> Ca3Al2Si3O12 | 1.44 (4)                | 1.69 (5324)    |
> Ca3Ti2O7         | 1.38 (2)                | 1.85 (3924)    |
> Cu28S16           | 1.97 (2)                | 2.22 (4708)    |
> K3Fe5F15         | 1.78 (2)                | 2.00 (5392)    |
> Y2O3                 | 1.45 (2)                | 2.22 (8218)    |
> SrTiO3              | 1.40 (4)                | 1.87 (3898)    |
> Sr2TiO4            | 1.41 (12)              | 1.84 (2850)    |
> Sr3Ti2O7          | 1.41 (8)                | 1.86 (2506)    |
> Sr4Ti3O10        | 1.41 (8)                | 1.86 (1950)    |
> -------------------------------------------------------------------------
>
> The shortest interatomic distances observed are plausible distances, the major observed peaks for each composition correspond to expected values when considering each chemistry. For example, the first peak at 1.87 Angstroms in SrTiO$_3$, is close to that of the Ti-O distances in the experimentally observed structure of 1.95 Angstroms. These observations in addition to the distribution of energies shown in Fig.12 of Appendix B.4 lead us to the conclusion that the final structures produced by MACS do not contain unphysically short interatomic distances that are typical for structures produced due to the failure of the energy calculator."

---

> > ### Comment · Reviewer_6w1f · 2025-08-04
> >
> > Thank you for your response, they have helped clarify some of the doubts I had.
> >
> > > The lack of novel methods that consistently outperform these baselines in general settings motivated our research
> >
> > > papers [1-3], the structure optimization method provided is either designed for a different task or tailored to more restrictive conditions than those addressed in our paper.
> >
> > Thanks for clearing this up. I still find it hard to assess the quality of the baselines: they are supported by the fact that MolOpt "exhibits performance comparable to FIRE, and is inferior to that of BFGS" (line 83). But MolOpt is not specifically designed for CSP either. The other work mentioned in related work ([2] in the references) does achieve lower energy compared to FIRE, but is also not designed for CSP. For example, [1] lists 13 CSP algorithms. I could not find any of them in the reference list. Also, [2] proposes a diffusion model for CSP. Are none of them relevant?
> >
> > As a disclaimer, I am not very familiar with the field, and am trying to get a better perspective on the related literature.
> >
> > > the final structures produced by MACS do not contain unphysically short interatomic distances
> >
> > Thank you for this analysis, it is very interesting! I believe it is does not substitute DFT, and there are other ways that MACS could have done to perform reward-hacking on CHGNet, but I appreciate the effort the authors put to provide additional insights on the discovered solutions.
> >
> > [1] Wei, L., Omee, S. S., Dong, R., Fu, N., Song, Y., Siriwardane, E., ... & Hu, J. (2024). CSPBench: a benchmark and critical evaluation of Crystal Structure Prediction. arXiv preprint arXiv:2407.00733.
> >
> > [2] Jiao, R., Huang, W., Lin, P., Han, J., Chen, P., Lu, Y., & Liu, Y. (2023). Crystal structure prediction by joint equivariant diffusion. Advances in Neural Information Processing Systems, 36, 17464-17497.

---

> > > ### Author Response · Authors · 2025-08-06
> > >
> > > Dear reviewer
> > >
> > > > Thanks for clearing this up. I still find it hard to assess the quality of the baselines: they are supported by the fact that MolOpt "exhibits performance comparable to FIRE, and is inferior to that of BFGS" (line 83). But MolOpt is not specifically designed for CSP either. The other work mentioned in related work ([2] in the references) does achieve lower energy compared to FIRE, but is also not designed for CSP. For example, [1] lists 13 CSP algorithms. I could not find any of them in the reference list. Also, [2] proposes a diffusion model for CSP. Are none of them relevant?
> > >
> > > The references [1, 2] are focused on crystal structure prediction (CSP), which is a global optimisation problem. MACS addresses geometry (local) optimization. This is a critical component of CSP, but is quite distinct from it. We can confirm that these CSP tools cannot serve as a benchmark for local optimisation.
> > >
> > > The task of CSP is to identify the global minimum structure on the potential energy surface. To do this a series of local geometry optimisations are performed starting from selected starting structures – MACS addresses the geometry optimisations only. These starting structures can be chosen from a pre-defined list of prototype structures, generated at random, or evolved using heuristics such as genetic algorithm, Bayesian optimisation, basin hoping etc... – it is the method of generating these starting structures that differentiates the different CSP tools. These structures are passed to an optimiser such as CG, BFGS or FIRE that uses an energy function such as DFT, MLIP or a forcefield to drive the geometry optimisation – we benchmark against these optimisers. We do not discuss how to navigate the energy landscape (the ultimate goal of the CSP tools in Ref. [1]) but how to optimise a given crystal structure (an integral part of any CSP exploration).
> > >
> > > As we state in Section 3.1: “**Geometry or local optimization** takes an initial structure as input and adjusts the positions of the atoms in the unit cell to achieve a structure where the energy is at a local minimum.” Regarding CSP, we note in Section 3.1: “The **global crystal structure optimization**, which is the ultimate goal of CSP, aims to identify the global minimum energy structure for a given composition, representing the most stable configuration. However, achieving a globally optimal structure typically requires many iterations of structure generation or perturbation followed by local optimization”.
> > >
> > > For instance, 6 out of 13 methods in [1] use DFT-calculated energy and most likely, optimize the structures using this energy (some of them may not optimize structures locally at all). In most of the cases, the calculations on the energy and the local optimization are performed using the very widely used VASP package, which offers three built-in optimization methods two of which (CG, MD) are present in some variations in our baselines, and the third one (RMM-DIIS) is tailored to structures which are already close to the global minimum and thus not comparable with MACS.
> > >
> > > ParetoCSP mentioned in [1] performs relaxation using the M3GNet interatomic potential, rather than with DFT. M3GNet in turn relies on optimization methods from the ASE package. Almost all of the optimization methods available in ASE are included in our baseline set. Indirectly, the CSP methods discussed in [1] support our claim that our set of baselines provides a comprehensive representation of geometry optimization methods used in practice.
> > >
> > > > Thank you for this analysis, it is very interesting! I believe it is does not substitute DFT, and there are other ways that MACS could have done to perform reward-hacking on CHGNet, but I appreciate the effort the authors put to provide additional insights on the discovered solutions.
> > >
> > > As noted above, geometry optimisation is performed using interatomic potentials rather than DFT by other researchers – here we are solely proposing to change the local optimiser to MACS. It is both correct and fair to state that evaluation of the reasonableness of interatomic distances is not the same as a DFT-based geometry optimisation. We feel that it does show that MACS with CHGNet as the energy calculator affords physically meaningful structures, consistent with CHGNet being trained on DFT energies and with MACS being a viable and distinct local optimiser.
> > >
> > > We would also like to note that minimum interatomic distance analysis is used to assess the validity of generated structures in [2]: “The structural valid rate is calculated as the percentage of the generated structures with all pairwise distances larger than 0.5 Å.” According to this definition, 100% of the final structures produced by MACS are valid. Moreover, the threshold of 0.5 Å can be increased by at least a factor of two without affecting this validity.x

---

> > > > ### Comment · Reviewer_6w1f · 2025-08-08
> > > >
> > > > Thank you for all your effort in the discussion, it has been quite insightful. To summarize:
> > > >
> > > > - my main concern was related to the (in my view) simplistic baselines. This discussion made me realize that the baseline seem appropriate, and thus properly support the benefit of MACS. It also showed my limited knowledge in the application field (as I come from an RL background).
> > > >
> > > > - my second concern focused on the appropriateness of evaluation. This was somewhat addressed by the authors (knowing that it is unreasonable to compute DFT calculations on all final structures during the rebuttal period).
> > > >
> > > > - my third concern was related to the independent nature of the agents in the MA system. Despite the fact that empirically, independent learners have been shown to perform well on SMAC, this does not mean that the MA target problem does not exist and does not affect the current application.
> > > >
> > > > In light of all of this, I will raise my score, as overall this work seems a solid advancement in a specific subfield of scientific discovery. I will also lower my confidence score, as I feel I am missing part of the necessary relevant literature.

---

### Official Review · Reviewer_ytGL · 2025-07-03

**Clarity:** 3
**Significance:** 4
**Originality:** 4
**Rating:** 5
**Confidence:** 4

**Summary:**

MACS (Multi-Agent Crystal-Structure optimization) frames periodic geometry relaxation as a partially observable Markov game. Each atom is an independent agent that observes itself and its 12 nearest neighbors, then outputs a 3-D displacement. Displacements are scaled by the atom’s current force norm; the reward is the log-drop in that norm. A single, shared policy is trained with independent Soft Actor-Critic and a replay buffer, using CHGNet for energy/force calls. Against six standard optimizers (BFGS, BFGS-LS, FIRE, FIRE+LS, CG, MDMin), MACS cuts mean optimization time (T_mean) by ~34 %, reduces the force-call budget (C_mean) by ~28 %, and posts the lowest failure rate (0.36 %) across 39 test sets, including zero-shot transfer to unseen compositions and larger cells.

**Questions:**

Questions:
1. Did you tune BFGS-LS and CG for the experiments?
2. Any preliminary results on treating lattice vectors as extra agents, as suggested in Limitations?

**Ethical Concerns:**

["NO or VERY MINOR ethics concerns only"]

**Final Justification:**

Overall Recommendation: 5 (Accept). The paper introduces a novel MARL formulation, per-atom agents under periodic boundary conditions, and demonstrates consistent improvements over strong classical optimizers across diverse test sets (including zero-shot and larger cells). The main limitations (minimal species encoding, fixed lattice vectors, reliance on CHGNet) are acknowledged. Presentation issues (a few typos, dense captions) are minor and easily fixed. Overall, the work is technically sound, original, and impactful.

**Limitations:**

The paper notes that atomic descriptors are minimal and unit-cell vectors are fixed/ it could also discuss how reliance on CHGNet accuracy affects performance and how the policy behaves if force evaluations are noisy.

**Quality:**

3

**Strengths And Weaknesses:**

**Strengths:**
- First MARL relaxer that treats every atom as an agent while respecting lattice periodicity.
- Consistent wins on 39 test sets spanning seen and unseen chemistries and 2 × larger cells.
- A single policy generalises to compositions and sizes never encountered during training.
- Detailed ablations on features, reward, and action scaling show which choices matter (Fig. 3, Table 2).
- Local observation graph, per-atom action scaling, and log-force reward are each justified by the ablations.
- Code is promised. The appendix lists hyper-parameters, standard errors, and per-test PF, supporting reproducibility.

**Weaknesses:**
- minor clarity: repeated “otpimization/ Evalation” typos; Table 1 caption dense; Fig 3 colours hard to read.

---

> ### Author Rebuttal · Authors · 2025-07-30
>
> **Summary Rebuttal**
>
> To avoid repetition, we have formulated a shared response for the reviewers addressing the questions and concerns raised by more than one reviewer.
>
> We believe that simplicity is a strength and have intentionally used the simplest MARL approach with independent learning. Instead of coordination via explicit communication, our method enables coordinated behavior through interactions among agents via interdependent rewards and overlapping local observations. This allows agents to adapt to each other’s behavior over time, resulting in implicit coordination. Another main reason is that independent methods tend to scale better to tasks involving a large number of agents and/or actions compared to centralized methods. This is crucial for our method’s scalability and applicability in computational chemistry, where the optimization of large structures is a major bottleneck in many computational methods.
>
> Our experiments demonstrate that, with a carefully designed setup of actions, observations, and rewards, independent learning can offer a significant improvement over state-of-the-art methods for crystal structure geometry optimization. We recognize the importance of convincing you that our baselines are strong enough for this claim to be valid, and we address this below.
>
> The baselines we use, although classical, still form the majority of methods available in widely used packages for computational chemistry, such as ASE, VASP, GULP, and CHGNet. These are employed in thousands of computational research projects. In short, our baselines represent the standard choices used daily by practitioners working in computational chemistry. The lack of novel methods that consistently outperform these baselines in general settings motivated our research. We appreciate the references provided by Reviewer 6w1f; however, as we explain below in our rebuttal to 6w1f, none of these are appropriate for our task.
>
> Finally, we would like to emphasize the contribution of our paper in formulating a challenging, physically grounded optimization environment for a ubiquitous task in condensed matter science, and in proposing an efficient, scalable MARL method that opens new directions for further research.
>
> **Rebuttal to ytGL**
>
> Dear Reviewer,
>
> Thank you for taking the time to read our paper and provide valuable feedback. We will fix the typos and improve Table 1 and Figure 3 as suggested. The answers to your questions are below.
>
> > Question 1: Did you tune BFGS-LS and CG for the experiments?
>
> **Answer 1**. All baselines were used with the default hyperparameters defined in the ASE package, as they represent a typical use case.
>
> > Question 2: Any preliminary results on treating lattice vectors as extra agents, as suggested in Limitations?
>
> **Answer 2**. This is an interesting future work, as mentioned in our paper, and preliminary results are not yet available.
>
> > Limitations: The paper notes that atomic descriptors are minimal and unit-cell vectors are fixed/ it could also discuss how reliance on CHGNet accuracy affects performance and how the policy behaves if force evaluations are noisy.
>
> **Answer 3**.
> We have conducted the additional analysis of the quality of the final structures produced by MACS, which is provided in the reply to reviewer 6w1f (Answer 4) and will be added to the Appendix. This analysis of the resulting structures reveals that they are physically acceptable.
> The noisy force evaluations would be an interesting direction for future research, and we will mention it in the revised paper.

---

### Decision · Program_Chairs · 2025-09-17

**Decision:**

Accept (poster)

**Comment:**

This paper proposes a multi-agent reinforcement learning (MARL) framework for crystal structure optimization, where each atom is represented by an independent SAC agent operating with a locally defined state space. The agents are trained with independent reward functions, enabling simultaneous learning across atoms. The method is evaluated against classical optimization techniques and demonstrates performance comparable to the best baselines while being significantly more efficient.

The reviewers acknowledged the technical soundness and originality of the proposed MARL formulation, particularly the novel use of per-atom agents under periodic boundary conditions. The empirical evaluation was considered robust, with consistent results across seeds and comparisons that fairly demonstrated the benefits of the method. While some initial concerns were raised regarding the strength of the baselines, the appropriateness of the evaluation protocol, and the limitations of independent learners, these were either adequately addressed in the rebuttal or found less critical upon further discussion.

That said, some limitations remain: the novelty is perceived as somewhat limited by certain reviewers, and design choices such as minimal species encoding, fixed lattice vectors, and reliance on CHGNet constrain the generality of the approach. Presentation issues (e.g., dense captions, minor typos) were also noted but are easily fixable. One reviewer also highlighted their reduced confidence in assessing the work due to limited familiarity with domain-specific literature. Another reviewer, with a rating of 3, is not convinced on the theoretical insights or guarantees provided on the benefits of compositionality, generalization bounds, or conditions under which MACS would outperform flat policies. The authors acknowledge this limitation. My personal belief as an AC is this limitation is not severe enough to undermine the other strengths of the work.

Overall, the paper is seen as a solid contribution to scientific discovery through MARL, advancing the state of the art in crystal structure optimization.